# SMC5/6-mediated plasmid silencing is directed by SIMC1–SLF2 and antagonized by the SV40 large T antigen

**Martina Oravcová[1], Minghua Nie[1], Takanori Otomo[2]\*, Michael N Boddy[1]\***

[1]Department of Molecular and Cellular Biology, The Scripps Research Institute, San Diego, United States; [2]San Diego Biomedical Research Institute, San Diego, United States

## eLife Assessment

This Research Advance manuscript further elucidates the roles of SMC5/6 loader proteins and associated factors in the silencing of extrachromosomal circular DNA by the SMC5/6 complex. While the findings are largely in line with expectations, they are **valuable**, representing a meaningful advance beyond the recent study from the same laboratories (PMC9708086), validating the previous model that distinct SMC5/6 subcomplexes, SIMC1-SLF2 and SLF1/2, separately control its transcriptional repression and DNA repair activities on extrachromosomal DNA. **Solid** evidence is presented for a role for SIMC1/SLF2 in localization of the SMC5/6 complex to plasmid DNA, and the distinct requirements as compared to recruitment of SMC5/6 to chromosomal DNA lesions.

**\*For correspondence:**
totomo@sdbri.org (TO);
nboddy@scripps.edu (MNB)

**Competing interest:** The authors declare that no competing interests exist.

## Abstract

SMC5/6 is unique among the Structural Maintenance of Chromosomes (SMC) complexes in its ability to repress transcription from extrachromosomal circular DNA (ecDNA), including viral genomes and plasmids. Previously, we showed that human SMC5/6 is regulated by two mutually exclusive subcomplexes—SIMC1–SLF2 and SLF1/2—the counterparts of yeast Nse5/6 (Oravcová et al., 2022). Notably, only SIMC1–SLF2 recruits SMC5/6 to SV40 large T antigen (LT) foci in PML nuclear bodies (PML NBs), suggesting that these regulatory subcomplexes direct distinct roles of SMC5/6 on chromosomal versus ecDNA. However, their roles in plasmid repression remain unclear. Here, we demonstrate that SMC5/6-mediated repression of plasmid transcription depends exclusively on SIMC1–SLF2, whereas SLF1/2 is dispensable. Reinforcing its specialized role in ecDNA suppression, SIMC1–SLF2 does not participate in SMC5/6 recruitment to chromosomal DNA lesions. We further show that plasmid silencing requires a conserved interaction between SIMC1–SLF2 and SMC6, mirroring the functional relationship observed between yeast Nse5/6 and Smc6. As for viral silencing, plasmid repression depends on the SUMO pathway; however, unlike viral silencing, it does not require PML NBs. Additionally, we find that LT interacts with SMC5/6 and increases plasmid transcription to levels observed in SIMC1–SLF2-deficient cells—echoing the antagonistic roles of HBx (HBV) and Vpr (HIV-1) in viral genome repression. These findings expand the paradigm of viral antagonism against SMC5/6-mediated silencing, positioning LT as a novel player in this evolutionary tug-of-war.

## Introduction

The Structural Maintenance of Chromosomes (SMC) complexes, present in both prokaryotes and eukaryotes, are core modulators of chromosome architecture and organization, supporting the development and homeostasis of all living organisms (*Hirano, 2006*; *Nasmyth and Haering, 2005*). The

SMC1/3 (cohesin) and SMC2/4 (condensin) complexes ensure accurate genome propagation by folding chromosomes into distinct structures. Meanwhile, SMC5/6 plays a key role in maintaining genome stability and integrity, contributing to processes such as homologous recombination (HR), alternative lengthening of telomeres (ALT), and ribosomal DNA (rDNA) replication (*Peng et al., 2018*; *Potts and Yu, 2007*; *Wehrkamp-Richter et al., 2012*; *Aragón, 2018*; *Räschle et al., 2015*). Recently, SMC5/6 has also been recognized as a viral restriction factor, highlighting its broader role in managing extrachromosomal circular DNAs (ecDNAs, *Irwan and Cullen, 2023*; *Peng and Zhao, 2023*; *Abdul et al., 2022*).

Beyond viral restriction, the repression of transcription from simple plasmid DNA has also been added to the activities of the human SMC5/6 'Swiss Army knife' of DNA manipulation (*Irwan and Cullen, 2023*; *Peng and Zhao, 2023*; *Abdul et al., 2022*; *Liu et al., 2023*; *Diman et al., 2024*). SMC5/6 selectively represses the transcription of circular but not linear plasmid DNA (*Diman et al., 2024*), which is likely related to their topological states. That is, DNA that accumulates superhelical stress during transcription is the preferred binding substrate for SMC5/6, whereas linear DNA that dissipates such tension is not stably bound by the complex (*Diman et al., 2024*; *Serrano et al., 2020*; *Jeppsson et al., 2024*; *Gutierrez-Escribano et al., 2020*). Once DNA is bound, SMC5/6 may further compact and thereby restrict transcriptional activity on the ecDNA.

All SMC complexes share a fundamental structure consisting of a SMC heterodimer connected by a kleisin protein. These complexes form a ring-like structure that traps DNA. The SMC proteins have a bent-arm shape and can open to encircle DNA when their ATPase head domains bind ATP, aiding DNA movement through the ring. Kleisin plays a critical role for connecting and locking the SMC heads together (*Aragón, 2018*; *Kim et al., 2023*; *Uhlmann, 2016*; *Adamus et al., 2020*; *Yu et al., 2021*; *Taschner et al., 2021*; *Li et al., 2024*; *Pradhan et al., 2023*). The SMC5/6 complex is particularly adept at recognizing and compacting special DNA structures, such as HR intermediates and supercoiled DNA (*Serrano et al., 2020*; *Gutierrez-Escribano et al., 2020*; *Palecek, 2018*; *Ryu et al., 2015*).

The SMC5/6 complex is composed of a highly conserved 'core' hexamer: the SMC5/6 heterodimer and non-SMC-elements (NSE) 1–4 (*Peng and Zhao, 2023*). A Kleisin (Nse4 in yeast; NSMCE4 in human) and two interacting partners (Nse1 and Nse3 in yeast; NSMCE1 and NSMCE3 in human) form a trimer that joins the SMC5/6 head domains. The SUMO ligase Nse2 (NSMCE2 in human) attaches to the SMC5 arm. The complex's functional specificity is further defined by non-core subunits, the Nse5/6 heterodimer (*Peng and Zhao, 2023*; *Adamus et al., 2020*; *Palecek et al., 2006*).

Yeast Nse5/6 plays key roles in regulating Smc5/6's genome stability functions, facilitating its chromatin loading, inhibiting its ATPase activity, and enhancing Nse2 SUMO ligase activity (*Yu et al., 2021*; *Bustard et al., 2016*; *Oravcová et al., 2019*; *Pebernard et al., 2006*; *Wan et al., 2019*; *Li et al., 2023*; *Hallett et al., 2021*; *Etheridge et al., 2021*). Despite its critical regulatory roles, the identification of a human Nse5/6-like complex lagged, largely due to the lack of cross-species protein sequence conservation. Nevertheless, identified in proteomic approaches, the human SLF1 (SMC5–SMC6 Complex Localization Factor 1)–SLF2 heterodimer (SLF1/2) was shown to fulfill some genome stability roles performed by Nse5/6 in yeast (*Räschle et al., 2015*). Interestingly, however, with the possible exception of HIV-1 proviral DNA, the repression of viral DNA transcription by SMC5/6 depends on SLF2 but not SLF1 (*Irwan and Cullen, 2023*; *Peng and Zhao, 2023*; *Liu et al., 2023*). Therefore, SLF2 either functions as a monomer in viral repression, or there is a cofactor that replaces SLF1 in this role. Toward resolving this issue, we recently discovered SIMC1 as a missing link in SMC5/6 regulation and demonstrated that the SIMC1–SLF2 complex is a human counterpart of yeast Nse5/6 (*Oravcová et al., 2022*).

Our structural and mutational analyses revealed that SIMC1 and SLF2 contain Nse5- and Nse6-like domains, respectively, which interact through a conserved interface. SIMC1–SLF2 interacts with and directs SMC5/6 to subnuclear compartments at PML nuclear bodies (PML NBs) induced by the SV40 polyomavirus large T antigen (LT). Further, we showed that SIMC1–SLF2 and SLF1/2 are mutually exclusive subcomplexes of SMC5/6 that likely support the distinct antiviral and DNA repair activities of the holocomplex. Thus, SIMC1–SLF2 and SLF1/2 act as Nse5/6-like regulatory elements that modulate the broad activities of the SMC5/6 complex (*Oravcová et al., 2022*).

Despite the foregoing advances, the specific roles of the SLF2-based SIMC1 and SLF1 subcomplexes in SMC5/6 regulation remained unclear. Although we previously proposed distinct functions

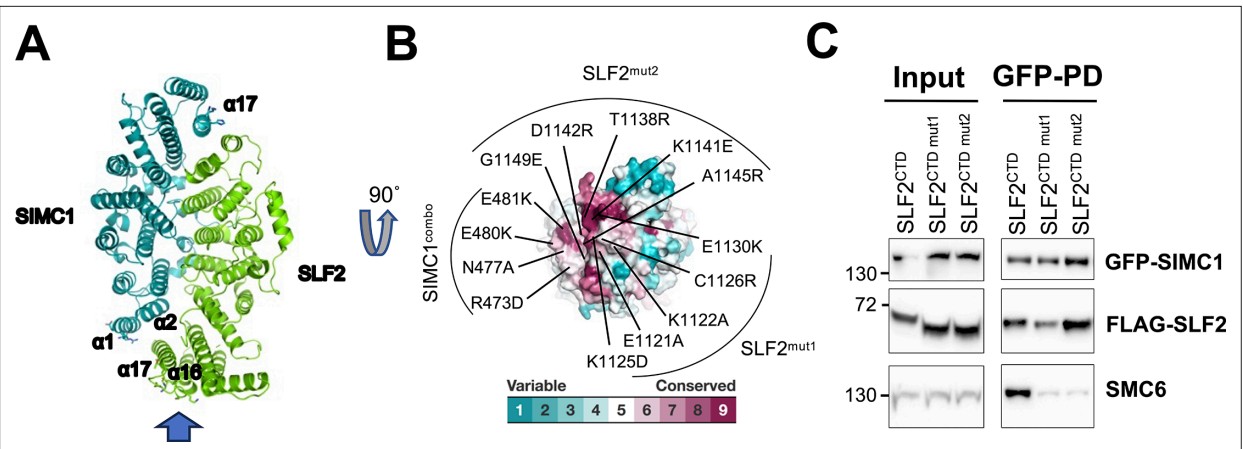

**Figure 1.** SIMC1–SLF2 composite surface patch is required for SMC6 association. (**A**) The cryo-EM structure of the SIMC1–SLF2 complex (PDB ID: 7T5P) (*Oravcová et al., 2022*). Residues mutated for interaction analysis are shown with side chains. The arrow at the bottom indicates the viewing angle for (**B**). (**B**) Conservation mapping on the surface of SIMC1–SLF2 complex, showing the N-terminus of SIMC1 and the C-terminus of SLF2. Conservation scores obtained from the Consurf server (*Ashkenazy et al., 2016*) are shown by the color graduation as indicated. Amino acids mutated in SIMC1$^{combo}$ (R473D/N477A/E480K/E481K in α1 of SIMC1), SLF2 mut1 (E1121A/K1122A/K1125D/C1126R/E1130K in α16 of SLF2), and SLF2 mut2 (T1138R/K1141E/D1142R/A1145R/G1149E in α17 of SLF2) are labeled. (**C**) Western blot of GFP-Trap immunoprecipitation from HEK293 cells transfected with plasmids expressing GFP-SIMC1, Myc-SMC6, and FLAG-tagged C-terminal domain (CTD) of SLF2, either WT or containing mutations (mut1, mut2). Input and GFP PD were detected with anti-GFP, FLAG, or SMC6 antibody. Full and unedited blots provided in *Figure 1—source data 1*.

The online version of this article includes the following source data and figure supplement(s) for figure 1:

**Source data 1.** Full and unedited blots corresponding to panel C.

**Figure supplement 1.** SIMC1 combo control mutant and SLF2$^{635-1160}$ retain interaction with SMC6.

**Figure supplement 1—source data 1.** Full and unedited blots corresponding to panel A.

**Figure supplement 1—source data 2.** Full and unedited blots corresponding to panel B.

for SIMC1–SLF2 and SLF1/2 in ecDNA silencing and DNA repair, respectively, these mechanisms had yet to be demonstrated. Additionally, SUMOylation, PML NBs, and SV40 LT were implicated in the regulation of SMC5/6-based ecDNA silencing, but their actual contributions remained untested. Here, we address these gaps directly. In line with our prior model, we show that SIMC1–SLF2, but not SLF1/2, mediates plasmid silencing through a direct and evolutionarily conserved interaction with SMC6. Conversely, SLF1/2 specifically recruits SMC5/6 to DNA lesions. Furthermore, we demonstrate that plasmid silencing is facilitated by the SUMO pathway but, unlike the SMC5/6-mediated silencing of HBV (*Niu et al., 2017*), does not require PML NBs. Lastly, we reveal that SV40 LT binds SMC5/6 and antagonizes plasmid silencing, thereby adding polyomaviruses to the growing list of pathogenic viruses that counteract SMC5/6-based transcriptional repression.

## Results

### SIMC1–SLF2 associates with SMC6 through a conserved patch

Our previous structural analysis revealed that the Nse5-like domain of SIMC1 (SIMC1$^{Nse5}$) and the Nse6-like domain of SLF2 (SLF2$^{Nse6}$) both adopt α-solenoid-like structures (*Oravcová et al., 2022*). These domains interact in a head-to-tail orientation, forming an ellipsoid-shaped dimer (*Figure 1A*). This dimerization clusters conserved residues from the N-terminus of SIMC1$^{Nse5}$ and the C-terminus of SLF2$^{Nse6}$ at one edge of the ellipsoid (*Figure 1B*). We previously used a SIMC1 mutant harboring four substitutions in α1 of SIMC1 (SIMC1$^{combo}$, *Figure 1B*) to demonstrate that this conserved patch of SIMC1 is essential for binding to SMC6 and for the localization of the SMC5/6 complex at LT-containing PML NBs (*Oravcová et al., 2022*).

In this study, we extended our analysis to SLF2 by introducing two sets of mutations into the Nse6-like region of SLF2 (SLF2$^{CTD}$; residues 635–1173): **mut 1** (E1121A/K1122A/K1125D/C1126R/E1130K in α16) and **mut 2** (T1138R/K1141E/D1142R/A1145R/G1149E in the subsequent loop and α17). These mutations were designed to disrupt charge interactions, hydrogen bonds, and van der

Waals contacts, following the same strategy used with SIMC1$^{combo}$ to disrupt binding (*Oravcová et al., 2022*). Using HEK293 cells co-expressing FLAG-tagged SLF2$^{CTD}$ mutants, GFP-tagged SIMC1, and Myc-tagged SMC6, we performed GFP pull-downs followed by Western blot analysis (*Figure 1C*). Wild-type SLF2$^{CTD}$ interacted with both SIMC1 and SMC6 (Lane 1), whereas both **mut 1** and **mut 2** disrupted SMC6 binding while preserving the interaction with SIMC1 (Lanes 2 and 3).

We also examined several control mutants. To confirm the SIMC1–SMC6 interface, we introduced a new SIMC1 control mutant containing four substitutions (Q842A/H846A/K849E/D857R) in α17, located at the opposite end of the ellipsoid, away from the predicted SMC6 contact surface (*Figure 1A*). This SIMC1 mutant bound to SLF2 but more weakly than WT SIMC1 (*Figure 1—figure supplement 1A*). This was an unexpected finding that is difficult to rationalize based on the structure. However, despite the reduced SLF2 association, it fully retained its interaction with SMC6, indicating that the residues mutated in SIMC1$^{combo}$ are specific to the SMC6 interface. In this experiment using overexpressed proteins, SIMC1 and SMC6 appear capable of maintaining their association during the pull-down even without stoichiometric SLF2. We then evaluated the most C-terminal residues of SLF2 (1161–1173), which are unstructured. Despite their conservation, deleting these residues (SLF2$^{635–1160}$) did not affect SMC6 binding (*Figure 1—figure supplement 1B*). Collectively, these results confirm that the conserved surface patch of SIMC1–SLF2 is essential for SMC6 binding.

## SIMC1–SLF2 subcomplex contacts the neck region of SMC6

We used AlphaFold-Multimer (*Evans et al., 2021*) to predict the structure of the SIMC1–SLF2–SMC6 complex (*Figure 2A*, *Figure 2—figure supplement 1A*). The resulting high-confidence model, as assessed by the predicted aligned error plot (*Figure 2—figure supplement 1B*), aligns well with mutational data and highlights the conserved surface patch of SIMC1–SLF2 as the interface with SMC6 (*Figure 2A*). SMC6 is a long, hairpin-like polypeptide in which its N- and C-termini fold together to form the ATPase 'head' domain at one end. The 'hinge' domain at the bending end mediates heterodimerization with SMC5 (*Figure 2—figure supplement 1A*). Connecting the head and hinge is a flexible coiled-coil; the portion adjacent to the head is referred to as the 'neck' (*Figure 2A*).

In the model, the conserved SIMC1–SLF2 surface patch interacts with C-terminal residues of SMC6, centered on the neck and opposite the ATPase active site where it dimerizes with SMC5's ATPase domain (*Li et al., 2024*; *Li et al., 2023*). This interface shows excellent shape complementarity—SMC6's convex neck helix fits snugly into the concave surface formed by α1 and α2 of SIMC1 and α17 of SLF2 (*Figure 2A*). Additional interactions arise from SLF2's α16 and the α16–α17 loop with SMC6's head residues, spanning a total buried surface area of 1516 Å².

This model explains the roles of the amino acids altered in our binding studies (*Figure 2B*; *Figure 2—figure supplement 1C, D*). The four residues modified in SIMC1$^{combo}$ all make direct contact with SMC6's neck helix, including three salt bridges: R473$^{SIMC1}$–D946$^{SMC6}$, E480$^{SIMC1}$–K957$^{SMC6}$/K942$^{SMC6}$, and E481$^{SIMC1}$–K942$^{SMC6}$. The SLF2 mutations also engage SMC6's neck region, forming salt bridges (E1130$^{SLF2}$–R940$^{SMC6}$ and K1141$^{SLF2}$–E1008$^{SMC6}$), a cation–π interaction (K1125$^{SLF2}$–Y944$^{SMC6}$), and multiple van der Waals contacts (*Figure 2A*; *Figure 2—figure supplement 1C, D*). Thus, the model and mutational data are consistent.

Next, we validated these SMC6 interface contacts using alanine substitutions. We introduced three sets of mutations: (1) **SIMC1-facing** residues (K942A/D946A/K957A), which form salt bridges with SIMC1; (2) **SLF2-facing** residues (R940A/Y944A/N947A), which form salt bridges and polar and hydrophobic contacts with SLF2; and (3) **SIMC1–SLF2-groove-facing** (L939A/R940A/K942A/L943A), which alters two leucines that insert into a hydrophobic groove between α1 of SIMC1 and α17 of SLF2, together with the SIMC1- and SLF2-facing residues (K942 and R940, respectively) (*Figure 2B*). When these SMC6 mutants were co-expressed with SIMC1 and SLF2, pull-down assays showed significantly reduced or abolished binding (*Figure 2C*, Lanes 2–4). Thus, our mutational data strongly support the AlphaFold model.

We conclude that SIMC1–SLF2 associates with the neck region of SMC6, opposite the ATPase active site used for heterodimerization with SMC5 (*Figure 2D*; *Figure 2—figure supplement 1A*). A recent cryo-EM study on the *S. cerevisiae* Smc5/6 complex similarly demonstrated that Nse5/6 engages with the neck region of Smc6 (*Li et al., 2023*), reinforcing the idea that this mode of interaction is evolutionarily conserved (*Figure 2D*). This has implications for the function of SIMC1–SLF2 in the regulation of SMC5/6 activity, based on the findings for yeast Nse5/6 (see discussion).

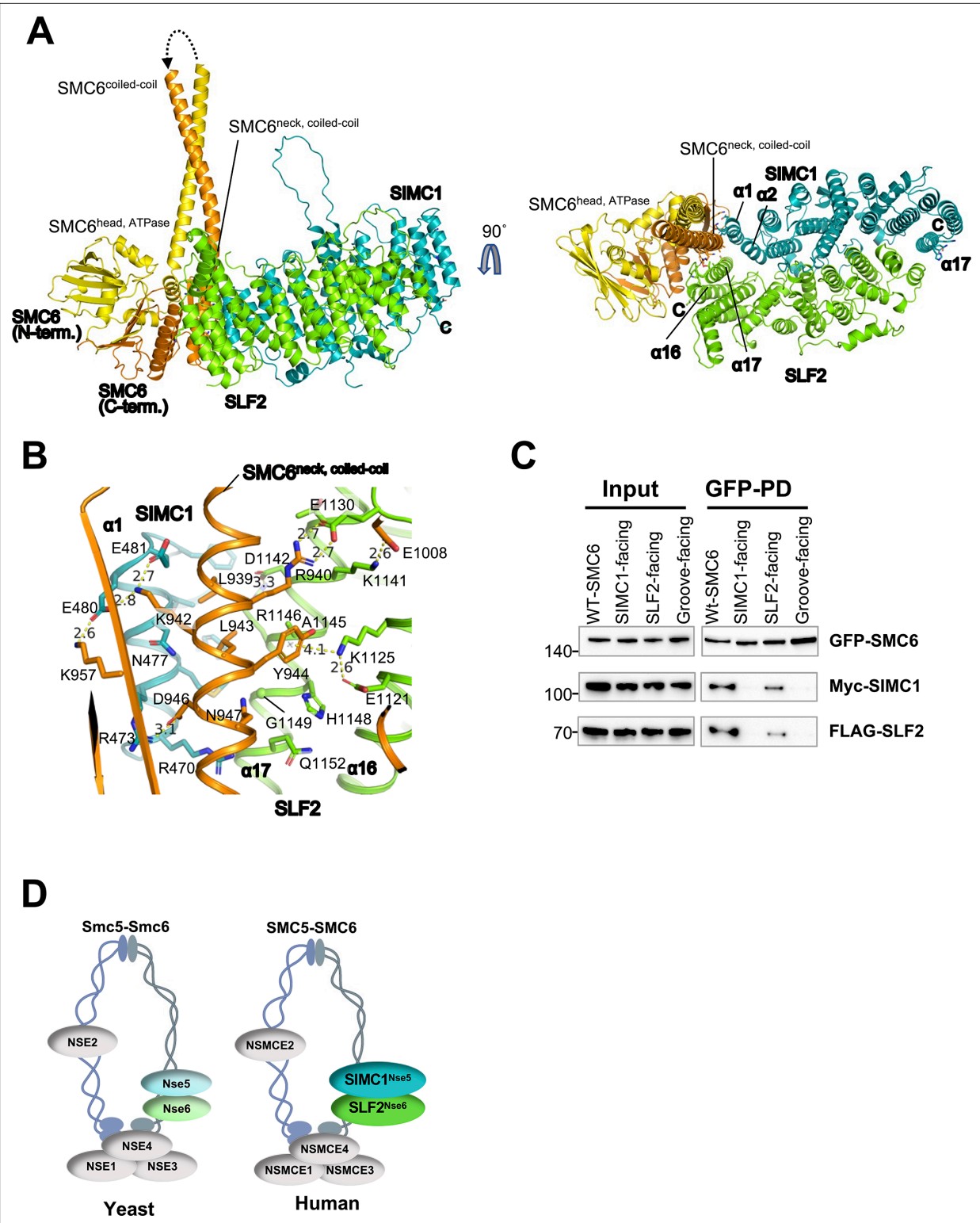

**Figure 2.** SIMC1–SLF2 subcomplex contacts the neck region of SMC6. (**A**) The AlphaFold-Multimer model of the SIMC1[Nse5]–SLF2[Nse6]–SMC6 complex. The disordered regions in the N- and C-termini, the hinge domain, and the majority of the coiled-coil that do not contact SIMC1–SLF2 are omitted for clarity. The entire model and the predicted aligned error (PAE) plot of the prediction are shown in *Figure 2—figure supplement 1*. Residues mutated for interaction analysis are shown with side chains. (**B**) A close-up view of the interface between SMC6's neck and the composite patch of SIMC1–SLF2. (**C**) Western blot of GFP-Trap immunoprecipitation from HEK293 cells transfected with plasmids expressing 2Myc-SIMC1, FLAG-SLF2[CTD], and WT or mutant GFP-SMC6 (SIMC1-facing: K942A, D946A, K957A; SLF2-facing: R940A, Y944A, N947A; groove-facing: L939A, R940A, K942A, L943A). Input and

*Figure 2 continued on next page*

*Figure 2 continued*

GFP PD were detected with anti-GFP, FLAG, or Myc antibody. Full and unedited blots provided in *Figure 2—source data 1*. (D) Schematic of yeast (left) and human (right) SMC5/6 complex illustrating positions of its subunits within the complex and Nse5/Nse6 and SIMC1/SLF2 cofactors, respectively, interacting with the neck region of SMC6.

The online version of this article includes the following source data and figure supplement(s) for figure 2:

**Source data 1.** Full and unedited blots corresponding to panel C.

**Figure supplement 1.** Structural model and interface analyses of the SMC6–SIMC1–SLF2 complex.

## SMC5/6-mediated plasmid silencing requires the Nse5/6-like SIMC1–SLF2 complex

To examine the role of SIMC1 and SLF2 in SMC5/6-mediated episomal DNA silencing, we used CRISPR–Cas9 to create U2OS SIMC1[−/−] or SLF2[−/−] cells. We first compared the expression of a transfected GFP reporter plasmid between WT and null cells. GFP transcripts, as quantified by RT-qPCR, showed a strong increase over WT in both SIMC1[−/−] and SLF2[−/−] cells (*Figure 3A*). This increase in GFP transcripts was not due to an increase in average plasmid copy number in SLF2 null cells, versus WT, as determined by qPCR (*Figure 3—figure supplement 1A*). We then used fluorescence-activated cell sorting (FACS) to monitor GFP expression (*Soboleski et al., 2005*) and found that both SIMC1[−/−] and

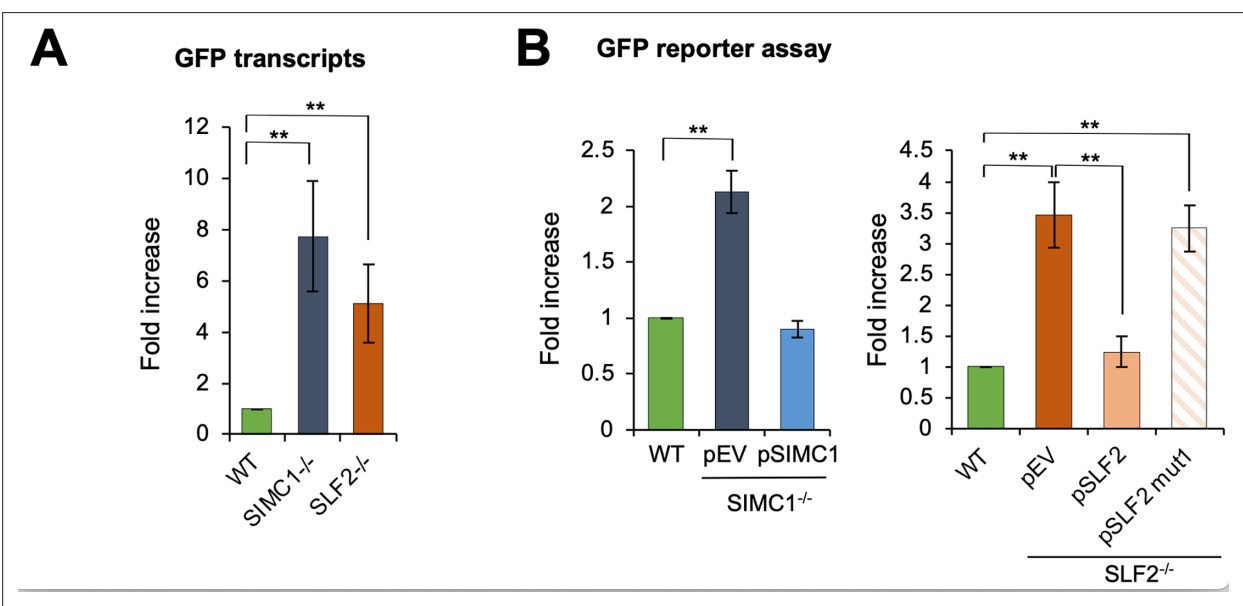

**Figure 3.** SIMC1/SLF2–SMC6 interaction is critical for plasmid silencing. (**A**) Expression of a GFP reporter transiently transfected into U2OS WT and SIMC1[−/−] or SLF2[−/−] cell lines was measured by reverse transcription-quantitative PCR 72 hr after transfection. GFP expression was normalized to the expression of beta-actin (mean ± s.d. from *n* = 4 independent experiments, two-tailed unpaired *t*-test; **p < 0.005). Primary data provided in *Figure 3— source data 1*. (**B**) U2OS WT, SIMC1[−/−], or SLF2[−/−] cells with integrated empty vector or vector expressing SIMC1 or SLF2 variants (WT, mut1), respectively, were transiently transfected with GFP reporter. After 72 hr, GFP intensity was measured by FACS and is displayed relative to GFP intensity measured in WT cells. Data are the average ± s.d. from *n* = 3 independent experiments, two-tailed unpaired *t*-test; **p < 0.005. No significant difference (p > 0.05) was found between WT and SIMC1[−/−] + pSIMC1 (left panel), or between WT and SLF2[−/−] + pSLF2 (right panel). Primary data provided in *Figure 3—source data 2*.

The online version of this article includes the following source data and figure supplement(s) for figure 3:

**Source data 1.** GFP transcripts qPCR corresponding to panel A.

**Source data 2.** GFP FACS corresponding to panel B.

**Figure supplement 1.** Plasmid silencing defect in SLF2[−/−] cells is not caused by increased plasmid copy number and is lost upon plasmid integration into the genome.

**Figure supplement 1—source data 1.** GFP DNA qPCR corresponding to panel A.

**Figure supplement 1—source data 2.** SLF2 transcripts qPCR corresponding to panel B.

**Figure supplement 1—source data 3.** GFP FACS corresponding to panel C.

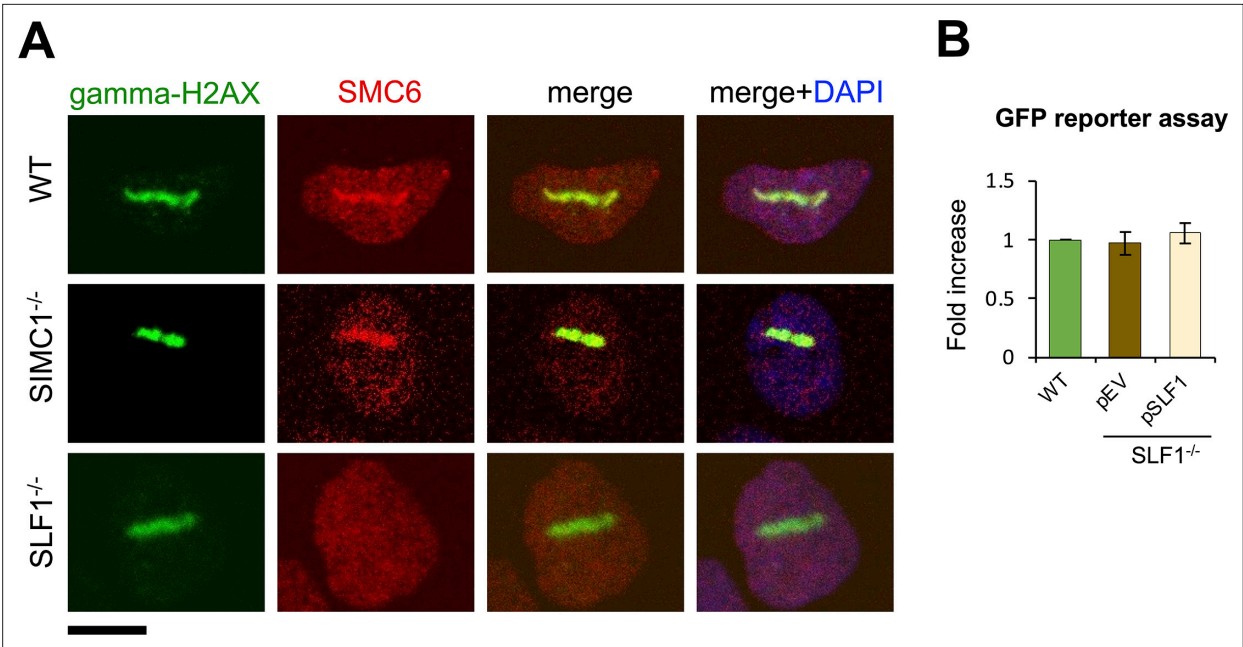

**Figure 4.** SLF1 is involved in SMC5/6 DNA damage repair, not plasmid silencing. (**A**) Representative immunofluorescence images of U2OS WT, SIMC1[-/-], SLF1[-/-] cells exposed to laser microirradiation upon treatment with 50 µM angelicin and 1 µg/ml Hoechst 33342 for 30 min. One hour after microirradiation, cells were pre-extracted, fixed, and stained with gamma-H2A.X (phosphorylation of histone H2A.X Ser139; green), SMC6 (red) antibodies and DAPI (blue). Scale bar 10 µm. (**B**) GFP intensity was measured by flow cytometry 72 hr after reporter transient transfection in U2OS WT cells or SLF1[-/-] cells with integrated empty vector or stably expressing SLF1. Data are the average ± s.d. from *n* = 3 independent experiments, two-tailed unpaired *t*-test (p-value >0.05). Primary data provided in *Figure 4—source data 1*.

The online version of this article includes the following source data for figure 4:

**Source data 1.** GFP FACS corresponding to panel B.

SLF2[-/-] cells expressed significantly more GFP than WT (*Figure 3B*). Importantly, expressing WT SIMC1 or SLF2 in the corresponding null cells reduced reporter expression to levels observed in WT cells (*Figure 3B*), demonstrating the involvement of SIMC1–SLF2 in plasmid silencing. Notably, however, SLF2[mut1] that abolishes the SIMC1–SLF2 interaction with SMC6 (*Figure 1C*) did not reduce reporter expression in SLF2[-/-] cells, despite being well expressed (*Figure 3—figure supplement 1B*). Importantly, in keeping with related studies (*Decorsière et al., 2016*; *Dupont et al., 2021*; *van Breugel et al., 2012*), we found that the effect of SLF2 on GFP expression was lost upon integration of the GFP reporter plasmid into the genome (*Figure 3—figure supplement 1C*). Together, these findings demonstrate that SIMC1–SLF2 and their interaction with SMC6 are required for the selective repression of episomal plasmid DNA.

## SIMC1 and SLF1 direct SMC5/6 to plasmid silencing and DNA repair, respectively

We previously showed that SLF1/2 and SIMC1–SLF2 form mutually exclusive complexes with SMC5/6 and recruit it to distinct cellular locations (*Oravcová et al., 2022*). Specifically, epitope-tagged SLF1 colocalized with gamma-H2AX at laser stripes, whereas SIMC1 did not. Here we more directly examined the roles of SLF1 and SIMC1 in recruiting SMC5/6 to DNA damage sites. To this end, we induced localized DNA lesions using laser microirradiation in WT, SLF1[-/-], and SIMC1[-/-] U2OS cells. In WT and SIMC1[-/-] cells, SMC5/6 accumulated at laser-induced lesions marked by gamma-H2AX, whereas no such accumulation was seen in SLF1[-/-] cells (*Figure 4A*). Thus, SLF1, but not SIMC1, plays a critical role in directing SMC5/6 to DNA damage sites. Conversely, when comparing GFP reporter expression in WT and SLF1 null cells, we found no change in GFP levels in the absence of SLF1 (*Figure 4B*). Thus, unlike SIMC1, SLF1 does not play a role in SMC5/6-mediated plasmid silencing, which is consistent with what has been observed in SLF1[-/-] HepG2 cells (*Abdul et al., 2022*).

## SUMOylation supports plasmid repression

Given that SIMC1 contains SUMO interacting motifs (SIMs), and the SMC5/6 complex is a SUMO ligase that contains other SUMO-interacting interfaces, we also tested for a potential involvement of the SUMO pathway in plasmid silencing (*Oravcová et al., 2022*; *Cho et al., 2024*; *Sun and Hunter, 2012*). Cells were treated with the SUMO pathway inhibitor (SUMOi), TAK-981 (*Langston et al., 2021*). We observed a strong dose-dependent increase in reporter gene expression in both U2OS and RPE cells (*Figure 5—figure supplement 1A, B*), supporting an involvement of the SUMO pathway in plasmid silencing. This is consistent with the broadly observed transcriptionally repressive activity of SUMOylation (*Boggio et al., 2004*; *Boulanger et al., 2021*). Interestingly, SUMOi treatment does not significantly increase the level of GFP reporter expression in SIMC1[-/-] and SLF2[-/-] over its effect on WT, despite the pre-existing defect in plasmid silencing in SIMC1 and SLF2 null cell lines (*Figure 5A*). The lack of additivity between SUMOi and SIMC1[-/-] or SLF2[-/-] suggests that SMC5/6-mediated plasmid silencing operates in large part under the broad 'umbrella' of SUMOylation-based transcriptional repression. Intriguingly, when the GFP reporter plasmid was integrated into the genome, its expression was unaffected by SUMOi treatment in either WT or SLF2[-/-] U2OS cells (*Figure 5—figure supplement 1C*). This was somewhat unexpected given the broadly inhibitory effect of SUMO on transcription. One possibility is that the dynamic range of transcriptional regulation is larger for episomal DNA, resulting in the muted response of the integrated reporter. This phenomenon merits further investigation.

## PML NBs are not required for plasmid silencing

We previously showed that SIMC1–SLF2 directs SMC5/6 to LT antigen-containing SV40 replication centers at PML NBs (*Oravcová et al., 2022*). Moreover, SMC5/6-mediated restriction of HBV occurs at PML NBs (*Niu et al., 2017*; *Yao et al., 2023*) and the localization of SMC5/6 at PML NBs is SLF2-dependent in HepG2 and PHH cells (*Abdul et al., 2022*; *Yao et al., 2023*). We also now find that SLF2 is required for the localization of SMC5/6 to PML NBs in U2OS cells (*Figure 5—figure supplement 2*). Based on the foregoing, PML NBs have been proposed, but not directly shown, to support simple plasmid repression (*Abdul et al., 2022*). Therefore, we tested the impact of PML NBs on plasmid silencing by measuring GFP reporter expression in WT and PML[-/-] U2OS cells (*Loe et al., 2020*). Compared to SIMC1 and SLF2 null cells, we observed only a minor increase in GFP expression in PML[-/-] versus WT cells (*Figure 5B*). Moreover, the stable re-expression of PML in PML[-/-] cells did not significantly reduce the level of GFP expression, as compared to vector control, suggesting the slight increase observed is due to clonal variation from CRISPR editing (*Figure 5B*). Thus, we conclude that PML NBs are not needed for the SMC5/6-mediated silencing of plasmids.

## SV40 LT antigen overcomes plasmid silencing

Considering that HBx and Vpr antagonize SMC5/6 to induce HBV and HIV-1 transcription, respectively (*Decorsière et al., 2016*; *Dupont et al., 2021*; *Murphy et al., 2016*), we tested if SV40 LT similarly impacts plasmid silencing. In this regard, we previously found that SMC5/6 colocalizes with LT-induced nuclear foci at PML NBs in a SIMC1–SLF2-dependent manner (*Oravcová et al., 2022*). Here, we show that ectopically expressed LT coimmunoprecipitates with GFP-SMC5/6 but not a GFP control (*Figure 5C*). Therefore, LT, like HBx and Vpr, may contact SMC5/6 to antagonize its ecDNA repression activity. Indeed, the stable expression of LT in WT cells promoted GFP reporter plasmid expression, as compared to WT cells with an empty vector control (*Figure 5D*; *Figure 5—figure supplement 3A*). The host range defective mutant LT-HR684 had a similar effect as WT LT, indicating that the C-terminus of LT, which inhibits the SV40 host range restriction factor FAM111A (*Fine et al., 2012*), is not involved in overcoming plasmid repression (*Figure 5D*). Notably, the enhancement of plasmid DNA transcription by LT is not additive with that already present in SIMC1[-/-] and SLF2[-/-] cells (*Figure 5D*). The epistatic relationship between the positive effects of SIMC1[-/-], SLF2[-/-], and LT expression on plasmid transcription indicates that LT overcomes SMC5/6-mediated repression. Interestingly, as we observed with SUMOi, when the GFP reporter plasmid was integrated in WT or SLF2[-/-] U2OS cells, there was not a pronounced effect of LT on GFP expression (*Figure 5—figure supplement 3B*). This is like the selective effect of HBx, whereby it overcomes the innate repression of transcription from episomal but not chromosomal reporters (*van Breugel et al., 2012*).

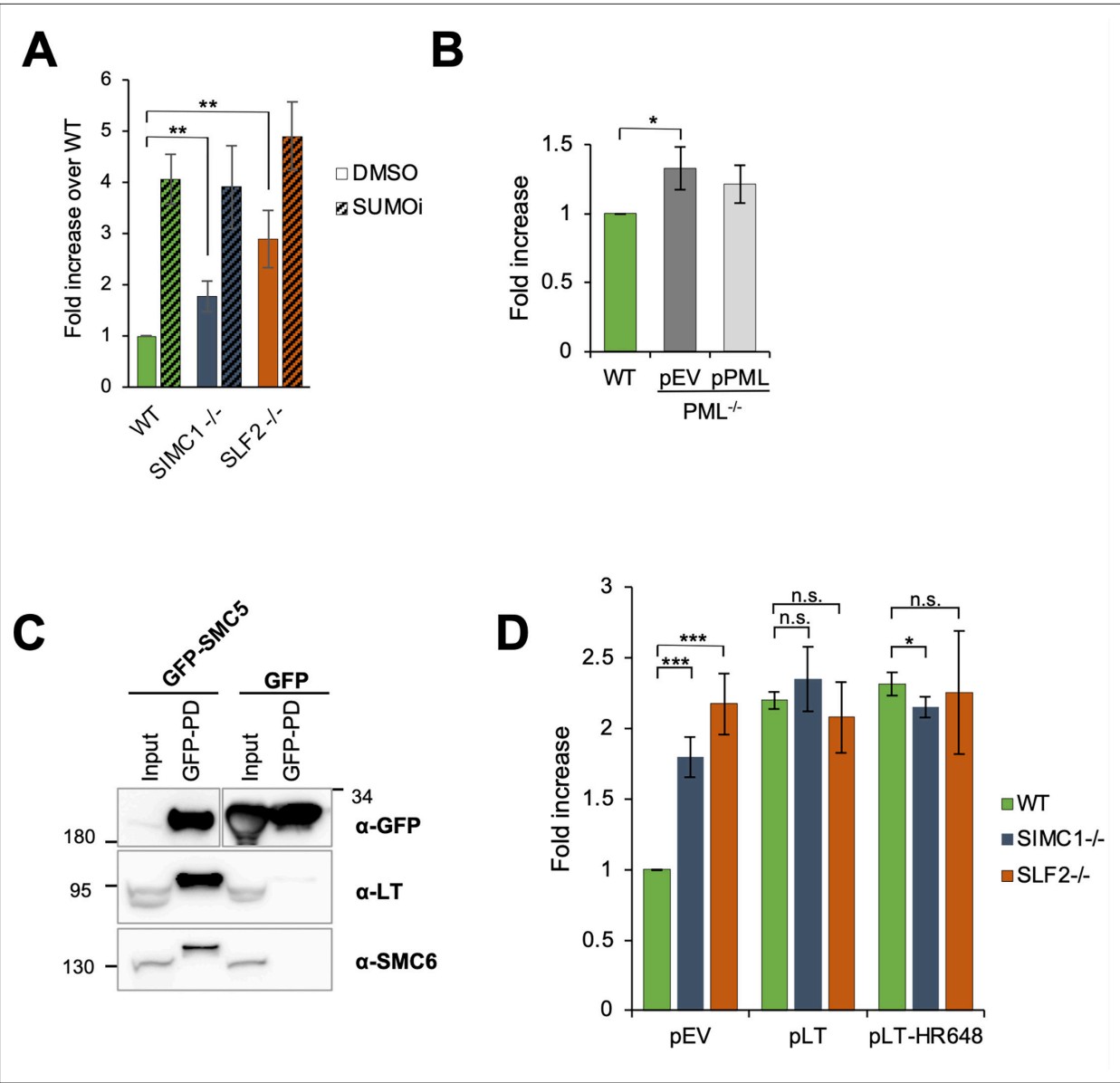

**Figure 5.** Plasmid silencing depends on the SUMO pathway and LT, not PML NBs. (**A**) U2OS WT, SIMC1[-/-], or SLF2[-/-] cells were transiently transfected with GFP reporter and treated with 100 nM TAK-981 (SUMOi) or DMSO at the same time. After 72 hr, GFP intensity was determined by FACS. Data are normalized to WT cells treated with DMSO and represent the average ± s.d. from $n = 4$ independent experiments, two-tailed unpaired $t$-test; **$p <$ 0.005. No significant difference was found between WT and SIMC1[-/-] or SLF2[-/-] when treated with SUMOi ($p > 0.05$). Primary data provided in *Figure 5— source data 1*. (**B**) U2OS WT or PML[-/-] cells with integrated empty vector or vector expressing PML, respectively, were transiently transfected with GFP reporter. Intensity of GFP was measured by FACS 72 hr after transfection. Data are the average ± s.d. from $n = 3$ independent experiments, two-tailed unpaired $t$-test; *$p < 0.05$. No significant difference ($p > 0.05$) was found between WT and PML[-/-] + pPML. Primary data provided in *Figure 5—source data 2*. (**C**) Western blot of GFP-trap immunoprecipitation from HEK293 cells transiently transfected with either GFP-SMC5 or GFP alone in combination with SV40 vector expressing LT and Myc-SMC6. Signals were visualized using GFP, LT, and SMC6 antibodies. Full and unedited blots provided in *Figure 5—source data 3*. (**D**) GFP intensity measured by FACS in U2OS WT, SIMC1[-/-], or SLF2[-/-] cells with integrated empty vector, vector expressing large T antigen (LT) or LT-HR684 variant, respectively, that were transiently transfected with GFP reporter for 72 hr. Data are the average ± s.d. from $n = 4$ independent experiments. *$p < 0.05$; ***$p < 0.0005$; n.s., $p > 0.05$ (two-tailed unpaired t-test). Primary data provided in *Figure 5—source data 4*.

The online version of this article includes the following source data and figure supplement(s) for figure 5:

**Source data 1.** GFP FACS corresponding to panel A.

**Source data 2.** GFP FACS corresponding to panel B.

**Source data 3.** Full and unedited blots corresponding to panel C.

**Source data 4.** GFP FACS corresponding to panel D.

*Figure 5 continued on next page*

*Figure 5 continued*

**Figure supplement 1.** Arbitrary luminescence units of U2OS (**A**) or RPE (**B**) cells expressing transfected pLuc reporter.

**Figure supplement 1—source data 1.** Luciferase measurement corresponding to panel A.

**Figure supplement 1—source data 2.** Luciferase measurement corresponding to panel B.

**Figure supplement 1—source data 3.** GFP FACS corresponding to panel C.

**Figure supplement 2.** Representative immunofluorescence images of U2OS WT or SLF2$^{-/-}$ cells fixed and stained with SMC6 (green) and PML (red) antibodies along with DAPI (blue).

**Figure supplement 3.** LT does not promote expression from integrated GFP reporter.

**Figure supplement 3—source data 1.** Full and unedited blots.

**Figure supplement 3—source data 2.** GFP FACS corresponding to panel B.

## Discussion

Our structural analyses indicate that SIMC1–SLF2 engages the SMC5/6 core at the neck region of SMC6, situated opposite the ATPase active site. These findings, together with recent cryo-EM analyses of the *S. cerevisiae* Smc5/6 complex (*Taschner et al., 2021*; *Li et al., 2024*; *Hallett et al., 2021*), highlight a conserved mode of interaction in which Nse5/6 or Nse5/6-like subunits bind to the backside of SMC6's head domain. This binding site is ideal for allosteric regulation of the ATPase and DNA-binding functions of the SMC5/6 family. Indeed, yeast Nse5/6 reduces Smc5/6 ATPase activity and disrupts the Nse4 (kleisin)–Smc6 interface, which may open the gate to enable topological DNA binding by Smc5/6 (*Figure 6*, *Taschner et al., 2021*; *Li et al., 2023*; *Hallett et al., 2021*). The clear overlap between our validated SIMC1–SLF2–SMC6 structure and that of yeast Nse5/6–Smc6 suggests that SIMC1–SLF2, and its orthologue SLF1/2, may similarly modulate SMC5/6 ATPase and DNA-binding activities, albeit at distinct DNA targets.

The distinct targeting functions of the SIMC1 and SLF1-based SLF2 complexes are underscored by our current analysis of microirradiated cells that lack each complex (*Figure 4*). While the deletion of SLF1 abrogates SMC5/6 localization at DNA lesions, the loss of SIMC1 has no effect (*Figure 4A*). On the other hand, SIMC1 is required for the recruitment of SMC5/6 to LT-containing PML NBs (*Oravcová et al., 2022*). Lastly, we found that SLF1 does not impact plasmid silencing (*Figure 4B*), consistent with several analyses of HIV-1, rAAV, and HBV silencing (*Abdul et al., 2022*; *Dupont et al., 2021*; *Yao et al., 2023*; *Ngo and Puschnik, 2023*) but contrasting with another study (*Irwan et al., 2022*).

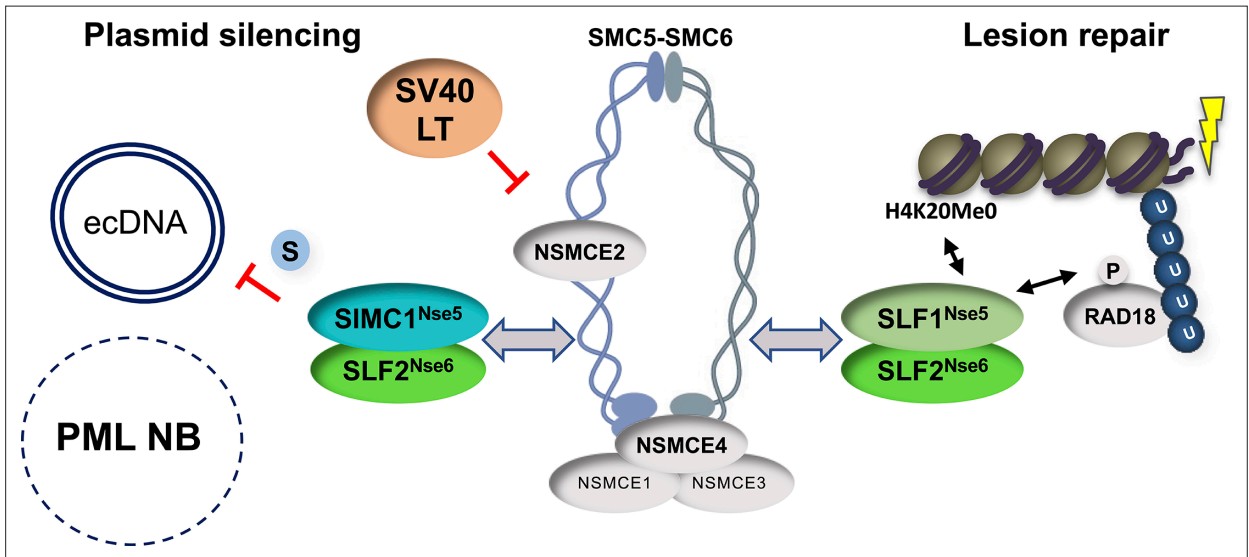

**Figure 6.** SIMC1–SLF2 and SLF1/2 subcomplexes direct SMC5/6 to separate pathways of plasmid silencing and DNA lesion repair. SIMC1–SLF2 and SMC5/6-mediated plasmid silencing is facilitated by the SUMO pathway and antagonized by SV40 LT. PML NBs are not involved in plasmid silencing. S stands for SUMO; U for ubiquitin. Both SIMC1–SLF2 and SLF1/2 likely contact SMC6 directly through their conserved Nse5/6-like domains, but the SIMC1 SIMs and SLF1 BRCT/ARD domains contact distinct posttranslational modifiers to direct the complex to its separate roles.

It remains unclear why this latter discrepancy exists. While there may still be overlapping functions of SIMC1 and SLF1 to uncover at the organismal level, it is clear they support distinct cellular responses to ecDNAs and DNA lesions (*Figure 6*).

In keeping with the SUMO-binding ability of SIMC1 (*Figure 6*, *Sun and Hunter, 2012*), we found that the inhibition of SUMOylation by TAK-981 strongly enhanced plasmid transcription (*Figure 5—figure supplement 2*). This is likely a broad effect that includes the loss of repressive modifications of transcription factors (*Boulanger et al., 2021*); as well as the lifting of SMC5/6-mediated repression to enhance ecDNA transcription. A similar effect of TAK-981 was seen on HIV-1 proviral DNA gene expression, which was elevated by SUMO inhibition to a similar degree as that observed in SMC5Δ cells (*Irwan et al., 2022*). The authors concluded that the effect of TAK-981 was entirely through the inhibition of SMC5/6-mediated SUMOylation (NSE2 subunit). This is different from our results, which show that SUMOylation inhibition has a larger effect than the loss of the SIMC1–SLF2 complex on plasmid gene expression. The distinct natures of the ecDNAs under study, plasmid versus HIV-1 proviral DNA, and the genetic backgrounds may contribute to this difference. Nevertheless, our data support a key role of the SUMO pathway in plasmid repression, and that SMC5/6 works within it.

Despite the fact that SIMC1–SLF2 localizes SMC5/6 to PML NBs that contain SV40 LT (*Hallett et al., 2021*), and HBV is restricted at PML NBs (*Niu et al., 2017*), they are dispensable for plasmid silencing.

While the reason for the different impact of PML NBs in HBV versus simple plasmid silencing remains unknown, it may be related to the natural localization of each ecDNA. That is, HBV localizes to PML NBs as part of the viral life cycle, in a manner dependent on the SUMOylation of its core protein (*Hofmann et al., 2023*). In contrast, the direct visualization of plasmids revealed they are pan-nuclear and apparently excluded from PML NBs (*Mearini et al., 2004*). It is also noteworthy that a recent genome-scale CRISPR screen for repressors of rAAV transcription identified all components of the SMC5/6 complex, including SIMC1 and SLF2, but not SLF1 or PML (*Ngo and Puschnik, 2023*). This raises interesting questions about the mechanism of SMC5/6-mediated transcriptional silencing, since it can occur outside PML NBs. In this regard, SUMOylation was recently shown to support the recombination-based ALT pathway in the absence of PML NBs (*Zhao et al., 2024*), which are normally key components of the fully active pathway (*Loe et al., 2020*). Therefore, the SUMOylation of plasmid-associated proteins and/or SMC5/6 may support transcriptional repression of plasmids in the absence of PML NBs, consistent with our findings using TAK-981.

Interestingly, the expression of LT enhances plasmid transcription in a non-additive manner with the deletion of SIMC1 or SLF2, suggesting an epistatic relationship. While LT is known to be a promiscuous transcriptional activator (*Sullivan and Pipas, 2002*; *Gilinger and Alwine, 1993*) that does not rule out a co-existing role in antagonizing SMC5/6. Indeed, these findings are reminiscent of HBx from HBV and Vpr of HIV-1, both of which are known promiscuous transcriptional activators that also directly antagonize SMC5/6 to relieve transcriptional repression (*Diman et al., 2024*; *Decorsière et al., 2016*; *Dupont et al., 2021*; *Murphy et al., 2016*; *Qadri et al., 1996*; *Aufiero and Schneider, 1990*; *Felzien et al., 1998*). Unlike HBx and Vpr, the interaction of LT with SMC5/6 does not appear to induce the degradation of its subunits (*Oravcová et al., 2022*). Nevertheless, LT binds and sterically inhibits several cellular proteins instead of inducing their degradation, including Rb and p53 (*Ahuja et al., 2005*). Further analysis will be needed to define LT-SMC5/6 contacts and its potential mode of inhibition. This will have important ramifications for the lifecycle of the pathogenic polyomaviruses since their LT proteins share functions with SV40 LT (*Googins et al., 2025*).

In conclusion, our structure–function analyses delineate a specific function for the SIMC1–SLF2 complex in SMC5/6-mediated plasmid repression. This resolves a discrepancy in the field, wherein SLF2 was thought to act alone in ecDNA repression, since SLF1 is not required. While further studies are needed, SIMC1–SLF2 is most likely responsible for supporting SMC5/6-mediated ecDNA silencing, including that of multiple pathogenic viruses. Although HBV silencing requires PML NBs, we find that plasmid silencing does not. This represents a departure from current dogma, suggesting that the SMC5/6-mediated repression mechanism does not always rely on PML NB-resident factors. In some cases, the SUMOylation of ecDNA-associated proteins may be sufficient to support SMC5/6-based repression. Finally, because the SV40 LT antigen interacts with SMC5/6 and antagonizes its repression of plasmids, the orthologous polyomaviral LT antigens are likely to block SMC5/6-mediated restriction of pathogenic polyomaviruses.

## Materials and methods
### Construction of recombinant plasmids
Construction of plasmid DNA was described in *Oravcová et al., 2022*. Mutations in SIMC1 (combo ctrl), SLF2 (mut1, mut2), and SMC6 (SIMC1-facing, SLF2-facing, SIMC1–SLF2-groove-facing) were either introduced in primers or a gBlock sequence containing desired mutations was purchased (IDT). All plasmids created in this study have been verified by sequencing service provided by Plasmid-saurus. Additional details of plasmid construction are available upon request.

### Cell culture, transfection, stable line generation
All cell lines and their derivatives were cultured in DMEM (Gibco, #11995065) supplemented with 10% (vol/vol) fetal bovine serum (Omega Scientific, #FB-01), 1% (vol/vol) antibiotic-antimycotic (Gibco, #15240062) and maintained at 37°C in humidified air with 5% $CO_2$.

Transient plasmid transfections and generation of stable cell lines were detailed in *Oravcová et al., 2022*. A list of cell lines and vectors used for generating stable cell lines or transient transfection is listed below:

| Designation | Source or reference | Identifier | Additional information |
|---|---|---|---|
| Cell line (Homo sapiens) | | | |
| HEK293 | https://doi.org/10.7554/eLife.79676 | | |
| HEK293T; U2OS | https://doi.org/10.7554/eLife.79676 | | |
| U2OS SIMC1⁻ᐟ⁻ | This paper | Clone B9 | Derived from U2OS by CRISPR/Cas9 |
| U2OS SLF1⁻ᐟ⁻ | This paper | Clone E9 | Derived from U2OS by CRISPR/Cas9 |
| U2OS SLF2⁻ᐟ⁻ | This paper | Clone 18 | Derived from U2OS by CRISPR/Cas9 |
| U2OS PML⁻ᐟ⁻ | https://doi.org/10.1101/gad.333963.119 | Clone 2C | |
| RPE | https://doi.org/10.1101/gad.333963.119 | | |
| Lentiviral vectors to generate stable cell line | | | |
| pHAGE2-FLAG-SIMC1 | https://doi.org/10.7554/eLife.79676 | pNB185 | |
| pHAGE2 | https://doi.org/10.7554/eLife.79676 | pNB248 | |
| pHAGE2-FLAG-SLF2 | This paper | pNB525 | |
| pHAGE2-FLAG-SLF2 mut1 | This paper | pNB527 | SLF2 mutations E1121A/K1122A/K1125D/C1126R/E1130K |
| pHAGE2-Flag-SLF1 | This paper | pNB555 | |
| pHAGE2-LT | This paper | pNB648 | |
| pHAGE2-GFP | This paper | pNB670 | |
| pHAGE2-LT-HR684 | This paper | pNB717 | |
| pMD2G | Addgene | #12259 | |
| psPAX2 | Addgene | #12260 | |
| Vectors used for transient transfection | | | |
| pDEST-eGFP-NLS-STOP | Helle Ulrich lab | pNB68, pNZ110 | |
| Dual secreted luciferase reporter | Addgene | #181934 pNB671 | EF1alpha-Gaussia luciferase, CMV-Cypridina luciferase |
| pBlueScript KS-SV40 | https://doi.org/10.1371/journal.ppat.1002949 | pNB371 | Strain 776 |
| pDEST-eGFP-SIMC1 | https://doi.org/10.7554/eLife.79676 | pNB263 | |
| pDEST-FLAG-NLS-SLF2 (635–1173) | https://doi.org/10.7554/eLife.79676 | pNB439 | SLF2^CTD |

*Continued on next page*

*Continued*

| Cell line (Homo sapiens) | | | |
|---|---|---|---|
| pDEST-GFP-SIMC1 combo1 mut | https://doi.org/10.7554/eLife.79676 | pNB497 | SIMC1 mutations R473D/N477A/E480K/E481K |
| pDEST-GFP-SIMC1 combo1 ctrl | This paper | pNB499 | SIMC1 mutations Q842A/H846A/K849E/D857R |
| pDEST-Myc-SMC6 | https://doi.org/10.7554/eLife.79676 | pNB530 | |
| pDEST-FLAG-NLS-SLF2 (635–1160) | This paper | pNB535 | |
| pDEST-FLAG-NLS-SLF2 (635–1173) mut1 | This paper | pNB536 | SLF2 mutations E1121A/K1122A/K1125D/C1126R/E1130K |
| pDEST-FLAG-NLS-SLF2 (635–1173) mut2 | This paper | pNB537 | SLF2 mutations T1138R/K1141E/D1142R/A1145R/G1149E |
| pDEST-GFP-SMC6 | This paper | pNB546 | |
| pDEST-GFP-SMC6 (groove-facing) | This paper | pNB677 | SMC6 mutations L939A/R940A/K942A/L943A |
| pDEST-GFP-SMC6 (SLF2-facing) | This paper | pNB678 | SMC6 mutations R940A/Y944A/N947A |
| pDEST-GFP-SMC6 (SIMC1-facing) | This paper | pNB679 | SMC6 mutations K942A/D946A/K957A |
| pDEST-2Myc-SIMC1 | This paper | pNB680 | |
| pDEST-Flag- LT wt | This paper | pNB739 | |

## Cell line generation by CRISPR/Cas9

SIMC1, SLF1, and SLF2 knockout clones were generated via reverse transfection of CRISPR/Cas9 Ribonucleoprotein. A mixture of three sgRNAs specific for each gene (Synthego; 24 nM) was combined with 6 nM Cas9-NLS (Synthego) in reduced-serum medium OptiMEM (Life Technologies, #31985062) and incubated for 10 min followed by a 15-min incubation with TransIT-X2 (Mirus, #MIR6000). $6 \times 10^4$ cells were added and plated into 12-well plates. After 72 hr, cells were diluted and plated to isolate single cell clones that were genotyped to confirm successful editing. sgRNA sequences and diagnostic primers used in this study:

| Target gene | Source | Identifier | Sequence 5'–3' |
|---|---|---|---|
| Gene knockout kit – SIMC1 | Synthego | Guides for human SIMC1 CRISPR/Cas9 | CCACAGGGACAAACTCTGCC; CTGCTGAAAGTCATCTTCTA; ATTGTGGGGCTGCTTGTCAC |
| SIMC1 CRISPR diagnostic_F | This paper | oNB576 | GGGCTTAGTATTTATGAGAGC |
| SIMC1 CRISPR diagnostic_R | This paper | oNB1023 | TGAATCACTGCACCTGGTCT |
| Gene knockout kit – SLF1 | Synthego | Guides for human SLF1 CRISPR/Cas9 | AGGAAAGTGGATACTAACCA; TTGATGAAACAACTTATGAA; CGTGAAGAACTGAAACGCAC |
| SLF1 CRISPR diagnostic_F | This paper | oNB1024 | TCCCCAAAATGCATAGTTCAAAG |
| SLF1 CRISPR diagnostic_R | This paper | oNB1025 | ACCATGGCTCATTTGGGCTA |
| Gene knockout kit – SLF2 | Synthego | Guides for human SLF2 CRISPR/Cas9 | ATGGTATACATGAGTCACGT; CTCCAAAAAGCAGACCACAG; AACTGGAATTTAGCTCCCAG |
| SLF2 CRISPR diagnostic_F | This paper | oNB1190 | CCCAAAAGGGTGCCACCAGA |

*Continued on next page*

*Continued*

| Target gene | Source | Identifier | Sequence 5′–3′ |
|---|---|---|---|
| SLF2 CRISPR diagnostic_R | This paper | oNB1191 | CTCCATTTGCTCCTTTCTCAACC |

## Co-immunoprecipitation and western blotting

GFP-labeled proteins were immunoprecipitated using the Nano-Trap magnetic agarose (Chromotek) following the manufacturer's instructions. The protocol as well as whole cell lysate preparation for SDS–PAGE and western blot is described in detail in *Oravcová et al., 2022*. Antibodies used for immunoblotting:

| Antibody | Source | Identifier | Dilution |
|---|---|---|---|
| GFP-Trap magnetic agarose (Alpaca Monoclonal) | ChromoTek | gtma | IP: 25 µl slurry |
| Myc-Trap magnetic agarose (Alpaca Monoclonal) | ChromoTek | ytma | IP: 25 µl slurry |
| Anti-FLAG (Mouse monoclonal) | Sigma | F3165 | 1:5000 |
| Anti-GFP (Mouse monoclonal) | Santa Cruz | sc-9996 | 1:10,000 |
| Anti-Myc (Mouse monoclonal) | Invitrogen | MA1-980 | 1:2000 |
| Anti-PSTAIR (Mouse monoclonal) | Sigma | P7962 | 1:8000 |
| Anti-SMC6 (Rabbit Polyclonal) | Bethyl | A300-237A | 1:1000 |
| Anti-SV40 LT (Mouse monoclonal) | Abcam | ab16879 | 1:5000 |
| Goat Anti-rabbit IgG, HRP (Goat Polyclonal) | Invitrogen | 31460 | 1:5000 |
| Goat Anti-mouse IgG, HRP (Goat Polyclonal) | Invitrogen | 31430 | 1:5000 |

## RNA extraction and quantitative PCR

For monitoring SLF2 and GFP expression by qPCR, cells were harvested for RNA isolation 72 hr after GFP vector transient transfection. Total RNA precipitated using RNeasy Plus Mini kit (QIAGEN, #74104) was treated with DNaseI kit (Invitrogen, #18068015) before cDNA synthesis by SuperScript III First-Strand Synthesis System for RT-PCR (Invitrogen, #18080-051). SensiFAST SYBR No-ROX kit (Meridian Bioscience, #BIO-98020) was used for qPCR and gene expression was calculated as $2^{-\Delta Ct}$ using beta-actin as a reference. Oligonucleotides used for RT-qPCR:

| Target gene | Source | Identifier | Sequence 5′–3′ |
|---|---|---|---|
| SLF2_F | This paper | oNB390 | GGAAGTCACTATTCATTC |
| SLF2_R | This paper | oNB476 | GAATTTAGCTCCCAGTGGG |
| GFP_F | This paper | oNB1220 | AGAAGAACGGCATCAAGG |
| GFP_R | This paper | oNB1221 | GCTCAGGTAGTGGTTGTC |
| beta actin_F | https://doi.org/10.7554/eLife.79676 | oNB492 | AGGCACCAGGGCGTGAT |
| beta actin_R | https://doi.org/10.7554/eLife.79676 | oNB493 | GCCCACATAGGAATCCTTCTGAC |

## DNA extraction for qPCR

U2OS cells ($1 \times 10^5$) with either integrated or transfected GFP vector (harvested 72 hr after GFP vector transfection) were resuspended in 100 µl PBS and lysed in 200 µl extraction buffer (0.1 M EDTA (pH 8.0), 0.5% (wt/vol) SDS, 10 mM Tris-Cl (pH 8.0)) supplemented with 40 µg Proteinase K, 5 µg RNaseA and incubated at 55°C for 1 hr, vortexed occasionally. Lysate was sonicated on Bioruptor Pico (Diagenode) in 10 cycles of 15 s ON and 30 s OFF and cleared by centrifugation at 18,000 × *g* 10 min. DNA was purified using NucleoSpin Gel and PCR clean-up kit (Macherey-Nagel,

#740609) with 5 volumes of NTB binding buffer based on manufacturer's instructions. Oligonucle-otides used for quantitative PCR:

| Target gene | Source | Identifier | Sequence 5'–3' |
|---|---|---|---|
| GFP_F | This paper | oNB1220 | AGAAGAACGGCATCAAGG |
| GFP_R | This paper | oNB1221 | GCTCAGGTAGTGGTTGTC |
| beta actin_F | This paper | oNB1378 | AGAGCAAGAGAGGCATCC |
| beta actin_R | This paper | oNB1379 | TTTCTCCATGTCGTCCCA |

## Flow cytometry

GFP fluorescence measurements were based on *Soboleski et al., 2005*. GFP reporter vector was transiently transfected into cells using TransIT-LT1 (Mirus, #MIR2300). After 72 hr, cells were washed in PBS, dissociated using 0.25% trypsin, and resuspended in DMEM. For SUMOi experiments, cells were treated with 100 nM TAK-981 (Cayman Chemical, #32741) or 0.1% DMSO (Sigma, #D8418) in control plates at the time of GFP vector transfection. Median GFP intensity was measured on Cytoflex S (Beckman Coulter) using the CytExpert software or on Aurora (Cytek) using SpectroFlo software. A minimum of 1100 GFP-positive cells were analyzed.

## DNA damage by laser microirradiation

Protocol adapted from *Tampere and Mortusewicz, 2016*. U2OS cells ($1.5 \times 10^5$) were plated in a 35-mm μ-Grid dish (Ibidi) a day before microirradiation. Cells were treated with 50 μM angelicin (Thermo Fisher, #501933704) and 1 μg/ml Hoechst 33342 (Invitrogen, #H3570) for 30 min before transferring to the stage incubator of LSM880 Airyscan confocal laser scanning microscope (Zeiss). Using the bleaching mode in ZEN software, DNA damage was induced by irradiation of a 5-pixel wide region with a 405-nm diode laser (15% power, 1 iteration, zoom 1, averaging 1, pixel dwell time 0.27 μs = speed 7). One hour after the irradiation, cells were treated with extraction buffer (0.5% Triton X-100, 20 mM HEPES, 50 mM NaCl, 3 mM $MgCl_2$, 300 mM sucrose) for 2 min, washed twice in PBS, fixed in 4% formaldehyde in PBS for 15 min and stained with anti-phospho-Histone H2A.X (Ser139) and anti-SMC6 antibodies following the immunofluorescence protocol.

## Immunofluorescence and microscopy

Immunofluorescence and confocal microscopy were done as described (*Oravcová et al., 2022*).

For optimal detection of SMC6, we pre-extracted the chromatin with extraction buffer (0.5% Triton X-100, 20 mM HEPES, 50 mM NaCl, 3 mM $MgCl_2$, 300 mM sucrose) for 2 min on ice. Antibodies used for immunofluorescence:

| Antibody | Source | Identifier | Dilution |
|---|---|---|---|
| Anti-phospho-Histone H2A.X (Ser139) (Mouse monoclonal) | Sigma | 05-636 | 1:500 |
| Anti-PML (Mouse monoclonal) | Santa Cruz | sc-966 | 1:200 |
| Anti-SMC6 (Rabbit Polyclonal) | Bethyl | A300-237A | 1:500 |
| Anti-SV40 LT (Mouse monoclonal) | Abcam | ab16879 | 1:400 |
| Goat Anti-mouse IgG (H+L), Alexa Fluor 488 (Goat Polyclonal) | Life Technologies | A11029 | 1:1000 |
| Goat Anti-rabbit IgG (H+L), Alexa Fluor 488 (Goat Polyclonal) | Life Technologies | A11008 | 1:1000 |
| Goat Anti-mouse IgG (H+L), Alexa Fluor 555 (Goat Polyclonal) | Life Technologies | A21422 | 1:1000 |
| Goat Anti-rabbit IgG (H+L), Alexa Fluor 555 (Goat Polyclonal) | Life Technologies | A21428 | 1:1000 |

## Luciferase and MTT assays

U2OS cells were seeded at $4 \times 10^3$ cells per well of a 96-well plate and transfected with 20 ng of luciferase reporter plasmid (expresses secreted Gaussian luciferase under EF1alpha promoter) using TransIT-LT1 (Mirus) by reverse transfection. TAK-981 in indicated concentrations or DMSO was added at the time of transfection, and Luciferase activities were measured 3 days later. Ten microliters of

culture media were mixed with 50 µl Gaussia Luciferase Glow Assay buffer that contains freshly diluted coelenterazine substrate (Pierce Gaussia Luciferase Glow Assay Kit, Thermo Fisher, #16160). Luminescence was measured after 10 min on a Tecan Infinite M200 plate reader (Tecan Life Sciences). The number of viable cells was estimated using the MTT assay kit (Sigma, #M2128) following the manufacturer's instructions.

## AlphaFold-Multimer structural modeling

AlphaFold-Multimer modeling (*Evans et al., 2021*) was performed using default settings on the following protein sequences: SIMC1[Nse5] (425–872), SLF2[Nse6] (733–1173), and SMC6 (1–1091).

## Material availability

All materials produced during this work are available upon written request, in keeping with the requirements of the journal, funding agencies, and The Scripps Research Institute.

## Acknowledgements

This work was supported by NIH grants R35 GM136273 to MNB and GM092740 to TO.

# Additional information

## Funding

| Funder | Grant reference number | Author |
| --- | --- | --- |
| National Institute of General Medical Sciences | R35 GM136273 | Martina Oravcová Minghua Nie Michael N Boddy |
| National Institute of General Medical Sciences | GM092740 | Takanori Otomo |

The funders had no role in study design, data collection, and interpretation, or the decision to submit the work for publication.

## Author contributions

Martina Oravcová, Minghua Nie, Formal analysis, Investigation, Methodology, Writing – review and editing; Takanori Otomo, Conceptualization, Formal analysis, Supervision, Funding acquisition, Investigation, Visualization, Methodology, Writing – original draft, Project administration, Writing – review and editing; Michael N Boddy, Conceptualization, Formal analysis, Supervision, Funding acquisition, Investigation, Methodology, Writing – original draft, Project administration, Writing – review and editing

## Author ORCIDs

Martina Oravcová ⓘ https://orcid.org/0000-0001-6063-2227
Takanori Otomo ⓘ https://orcid.org/0000-0003-3589-238X
Michael N Boddy ⓘ https://orcid.org/0000-0001-7618-4449

Reviewer #1 (Public review): https://doi.org/10.7554/eLife.106815.3.sa1
Reviewer #2 (Public review): https://doi.org/10.7554/eLife.106815.3.sa2
Reviewer #3 (Public review): https://doi.org/10.7554/eLife.106815.3.sa3
Author response https://doi.org/10.7554/eLife.106815.3.sa4

# Additional files

## Supplementary files

MDAR checklist

## Data availability

Source Data for all figures is uploaded.

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
