## [Editor Report · eLife Assessment]

This Research Advance manuscript further elucidates the roles of SMC5/6 loader proteins and associated factors in the silencing of extrachromosomal circular DNA by the SMC5/6 complex. While the findings are largely in line with expectations, they are **valuable**, representing a meaningful advance beyond the recent study from the same laboratories (PMC9708086), validating the previous model that distinct SMC5/6 subcomplexes, SIMC1-SLF2 and SLF1/2, separately control its transcriptional repression and DNA repair activities on extrachromosomal DNA. **Solid** evidence is presented for a role for SIMC1/SLF2 in localization of the SMC5/6 complex to plasmid DNA, and the distinct requirements as compared to recruitment of SMC5/6 to chromosomal DNA lesions.

---

## [Referee Report · Reviewer #1 (Public review)]

SMC5/6 is a highly conserved complex able to dynamically alter chromatin structure, playing in this way critical roles in genome stability and integrity that include homologous recombination and telomere maintenance. In the last years, a number of studies have revealed the importance of SMC5/6 in restricting viral expression, which is in part related to its ability to repress transcription from circular DNA. In this context, Oravcova and colleagues recently reported how SMC5/6 is recruited by two mutually exclusive complexes (orthologs of yeast Nse5/6) to SV40 LT-induced PML nuclear bodies (SIMC/SLF2) and DNA lesions (SLF1/2). In this current work, the authors extend this study, providing some new results.

---

## [Referee Report · Reviewer #2 (Public review)]

Oracová et al. present data supporting a role for SIMC1/SLF2 in silencing plasmid DNA via the SMC5/6 complex. Their findings are of interest, and they provide further mechanistic detail of how the SMC5/6 complex is recruited to disparate DNA elements. In essence, the present report builds on the author's previous paper in eLife in 2022 (PMID: 36373674, "The Nse5/6-like SIMC1-SLF2 complex localizes SMC5/6 to viral replication centers") by showing the role of SIMC1/SLF2 in localisation of the SMC5/6 complex to plasmid DNA, and the distinct requirements as compared to recruitment to DNA damage foci.

---

## [Referee Report · Reviewer #3 (Public review)]

This study by the Boddy and Otomo laboratories further characterizes the roles of SMC5/6 loader proteins and related factors in SMC5/6-mediated repression of extrachromosomal circular DNA. The work shows that mutations engineered at an AlphaFold-predicted protein-protein interface formed between the loader SLF2/SIMC1 and SMC6 (similar to the interface in the yeast counterparts observed by cryo-EM) prevent co-IP of the respective proteins. The mutations in SLF2 also hinder plasmid DNA silencing when expressed in SLF2-/- cell lines, suggesting that this interface is needed for silencing. SIMC1 is dispensable for recruitment of SMC5/6 to sites of DNA damage, while SLF1 is required, thus separating the functions of the two loader complexes. Preventing SUMOylation (with a chemical inhibitor) increases transcription from plasmids but does not in SLF2-deleted cell lines, indicating the SMC5/6 silences plasmids in a SUMOylation dependent manner. Expression of LT is sufficient for increased expression, and again, not additive or synergistic with SIMC1 or SLF2 deletion, indicating that LT prevents silencing by directly inhibiting 5/6. In contrast, PML bodies appear dispensable for plasmid silencing.

---

## [Author Response]

The following is the authors’ response to the original reviews

**Reviewer #1 (Public review):**
SMC5/6 is a highly conserved complex able to dynamically alter chromatin structure, playing in this way critical roles in genome stability and integrity that include homologous recombination and telomere maintenance. In the last years, a number of studies have revealed the importance of SMC5/6 in restricting viral expression, which is in part related to its ability to repress transcription from circular DNA. In this context, Oravcova and colleagues recently reported how SMC5/6 is recruited by two mutually exclusive complexes (orthologs of yeast Nse5/6) to SV40 LT-induced PML nuclear bodies (SIMC/SLF2) and DNA lesions (SLF1/2). In this current work, the authors extend this study, providing some new results. However, as a whole, the story lacks unity and does not delve into the molecular mechanisms responsible for the silencing process. One has the feeling that the story is somewhat incomplete, putting together not directly connected results.

Please see the introductory overview above.

(1) In the first part of the work, the authors confirm previous conclusions about the relevance of a conserved domain defined by the interaction of SIMC and SLF2 for their binding to SMC6, and extend the structural analysis to the modelling of the SIMC/SLF2/SMC complex by AlphaFold. Their data support a model where this conserved surface of SIMC/SLF2 interacts with SMC at the backside of SMC6's head domain, confirming the relevance of this interaction site with specific mutations. These results are interesting but confirmatory of a previous and more complete structural analysis in yeast (Li et al. NSMB 2024). In any case, they reveal the conservation of the interaction. My major concern is the lack of connection with the rest of the article. This structure does not help to understand the process of transcriptional silencing reported later beyond its relevance to recruit SMC5/6 to its targets, which was already demonstrated in the previous study.

Demonstrating the existence of a conserved interface between the Nse5/6-like complexes and SMC6 in both yeast and human is foundationally important, not confirmatory, and was not revealed in our previous study. It remains unclear how this interface regulates SMC5/6 function, but yeast studies suggest a potential role in inhibiting the SMC5/6 ATPase cycle. Nevertheless, the precise function of Nse5/6 and its human orthologs in SMC5/6 regulation remain undefined, largely due to technical limitations in available in vivo analyses. The SIMC1/SLF2/SMC6 complex structure likely extends to the SLF1/2/SMC6 complex, suggesting a unifying function of the Nse5/6-like complexes in SMC5/6 regulation, albeit in the distinct processes of ecDNA silencing and DNA repair. There have been no studies to date (including this one) showing that SIMC1-SLF2 is required for SMC5/6 recruitment to ecDNA. Our previous study showed that SIMC1 was needed for SMC5/6 to colocalize with SV40 LT antigen at PML NBs. Here we show that SIMC1 is required for ecDNA repression, in the absence of PML NBs, which was not anticipated.

(2) In the second part of the work, the authors focus on the functionality of the different complexes. The authors demonstrate that SMC5/6's role in transcription silencing is specific to its interaction with SIMC/SLF2, whereas SMC5/6's role in DNA repair depends on SLF1/2. These results are quite expected according to previous results. The authors already demonstrated that SLF1/2, but not SIMC/SLF2, are recruited to DNA lesions. Accordingly, they observe here that SMC5/6 recruitment to DNA lesions requires SLF1/2 but not SIMC/SLF2. Likewise, the authors already demonstrated that SIMC/SLF2, but not SLF1/2, targets SMC5/6 to PML NBs. Taking into account the evidence that connects SMC5/6's viral resistance at PML NBs with transcription repression, the observed requirement of SIMC/SLF2 but not SLF1/2 in plasmid silencing is somehow expected. This does not mean the expectation has not to be experimentally confirmed. However, the study falls short in advancing the mechanistic process, despite some interesting results as the dispensability of the PML NBs or the antagonistic role of the SV40 large T antigen. It had been interesting to explore how LT overcomes SMC5/6-mediated repression: Does LT prevent SIMC/SLF2 from interacting with SMC5/6? Or does it prevent SMC5/6 from binding the plasmid? Is the transcription-dependent plasmid topology altered in cells lacking SIMC/SLF2? And in cells expressing LT? In its current form, the study is confirmatory and preliminary. In agreement with this, the cartoons modelling results here and in the previous work look basically the same.

Our previous study only examined the localization of SLF1 and SIMC1 at DNA lesions. The localization of these subcomplexes alone should not be used to define their roles in SMC5/6 localization. Indeed, the field is split in terms of whether Nse5/6-like complexes are required for ecDNA binding/loading, or regulation of SMC5/6 once bound.

We agree, determining the potential mechanism of action of LT in overcoming SMC5/6-based repression is an important next step. We believe it is unlikely due to blocking of the SMC5/6SIMC1/SLF2 interface, since SIMC1-SLF2 is required for SMC5/6 to localize at LT-induced foci. It will require the identification of any direct interactions with SMC5/6 subunits, and better methods for assessing SMC5/6 loading and activity on ecDNAs. Unlike HBx, Vpr, and BNRF1 it does not appear to induce degradation of SMC5/6, making it a more complex and interesting challenge. Also, the dispensability of PML NBs in plasmid silencing versus viral silencing raises multiple important questions about SMC5/6’s repression mechanism.

(3) There are some points about the presented data that need to be clarified.

Thank you, we have addressed these points below, within the Recommendations for authors section.

**Reviewer #2 (Public review):**
Oracová et al. present data supporting a role for SIMC1/SLF2 in silencing plasmid DNA via the SMC5/6 complex. Their findings are of interest, and they provide further mechanistic detail of how the SMC5/6 complex is recruited to disparate DNA elements. In essence, the present report builds on the author's previous paper in eLife in 2022 (PMID: 36373674, "The Nse5/6-like SIMC1-SLF2 complex localizes SMC5/6 to viral replication centers") by showing the role of SIMC1/SLF2 in localisation of the SMC5/6 complex to plasmid DNA, and the distinct requirements as compared to recruitment to DNA damage foci. Although the findings of the manuscript are of interest, we are not yet convinced that the new data presented here represents a compelling new body of work and would better fit the format of a "research advance" article. In their previous paper, Oracová et al. show that the recruitment of SMC5/6 to SV40 replication centres is dependent on SIMC1, and specifically, that it is dependent on SIMC1 residues adjacent to neighbouring SLF2.

We agree. We submitted this manuscript as a “Research Advance”, not as a standalone research article, given that it is an extension of our previous “Research Article” (1).

Other comments(1) The mutations chosen in Figure 1 are quite extensive - 5 amino acids per mutant. In addition, they are in many cases 'opposite' changes, e.g., positive charge to negative charge. Is the effect lost if single mutations to an alanine are made?

The mutations were chosen to test and validate the predicted SIMC1-SLF2-SMC6 structure i.e. the contact point between the conserved patch of SIMC1-SLF2 and SMC6. Multiple mutations and charge inversions increased the chance of disrupting the extensive interface. In this respect, the mutations were successful and informative, confirming the requirement of this region in specifically contacting SMC6. Whilst alanine scanning mutations are possible, we believe that they would not add to, or detract from, our validation of the predicted SIMC1-SLF2-SMC6 interface.

(2) In Figure 2c, it isn't clear from the data shown that the 'SLF2-only' mutations in SMC6 result in a substantial reduction in SIMC1/SLF2 binding.

To clarify the difference between wild-type and SLF2-only mutations in SIMC1-SLF2 interaction, we have performed an image volume analysis. This shows that the SLF2-facing SMC6 mutant reduces its interaction with SIMC1 (to 44% of WT) and SLF2 (to 21% of WT). The reduction in both SIMC1 and SLF2 interaction with SMC6 SLF2-facing mutant is expected, since SIMC1 and SLF2 are an interdependent heterodimer.

**Author response table 1. sa4table1:** 

mutant	%
GFP-SMC6 blot	
wt-SMC6	100
SLF2 only	120,0891
Myc-SIMC1 blot	
wt-SMC6	100
SLF2 only	43,88612
Flag-SLF2 blot	
wt-SMC6	100
SLF2 only	21,28111

(3) In the GFP reporter assays (e.g. Figure 3), median fluorescence is reported - was there any observed difference in the percentage of cells that are GFP positive?

Yes, as expected when the GFP plasmid is not actively repressed, the percent of GFP positive cells differs in each cell line – in the same trend as GFP intensity

(4) The potential role of the large T antigen as an SMC5/6 evasion factor is intriguing. However, given the role of the large T antigen as a transcriptional activator, caution is required when interpreting enhanced GFP fluorescence. Antagonism of the SMC5/6 complex in this context might be further supported by ChIP experiments in the presence or absence of large T. Can large T functionally substitute for HBx or HIV-Vpr?

We agree, the potential role of LT in SMC5/6 antagonism is interesting. We did state in the text “While LT is known to be a promiscuous transcriptional activator (2,3) that does not rule out a co-existing role in antagonizing SMC5/6. Indeed, these findings are reminiscent of HBx from HBV and Vpr of HIV-1, both of which are known promiscuous transcriptional activators that also directly antagonize SMC5/6 to relieve transcriptional repression (4-10).“ We have tried ChIP experiments, but found these to be unreliable in assessing SMC5/6 association with plasmid DNA. Given the many disparate targets of LT, HBx and Vpr (other than SMC5/6), it seems unlikely that LT could functionally substitute for HBx and Vpr in supporting HBV and HIV-1 infections. Whilst certainly an interesting future question, we believe it is beyond the scope of this study.

(5) In Figure 5c, the apparent molecular weight of large T and SMC6 appears to change following transfection of GFP-SMC5 - is there a reason for this?

We are not certain as to what causes the molecular weight shift, but it is not specifically related to GFPSMC5 transfection. Rather, it appears to be a general effect of the pulldown. Indeed, a very weak “background” band of LT is seen in the GFP only pulldown, which also runs at a “higher” molecular weight, as in the GFP-SMC5 pulldown. We believe that the effect is instead related to gel mobility in the wells that contain post pulldown proteins and different buffers. We have also seen similar effects using different protein-protein interaction pairs.

**Reviewer #3 (Public review):**
Summary:This study by the Boddy and Otomo laboratories further characterizes the roles of SMC5/6 loader proteins and related factors in SMC5/6-mediated repression of extrachromosomal circular DNA. The work shows that mutations engineered at an AlphaFold-predicted protein-protein interface formed between the loader SLF2/SIMC1 and SMC6 (similar to the interface in the yeast counterparts observed by cryo-EM) prevent co-IP of the respective proteins. The mutations in SLF2 also hinder plasmid DNA silencing when expressed in SLF2-/- cell lines, suggesting that this interface is needed for silencing. SIMC1 is dispensable for recruitment of SMC5/6 to sites of DNA damage, while SLF1 is required, thus separating the functions of the two loader complexes. Preventing SUMOylation (with a chemical inhibitor) increases transcription from plasmids but does not in SLF2-deleted cell lines, indicating the SMC5/6 silences plasmids in a SUMOylation dependent manner. Expression of LT is sufficient for increased expression, and again, not additive or synergistic with SIMC1 or SLF2 deletion, indicating that LT prevents silencing by directly inhibiting 5/6. In contrast, PML bodies appear dispensable for plasmid silencing.Strengths:The manuscript defines the requirements for plasmid silencing by SMC5/6 (an interaction of Smc6 with the loader complex SLF2/SIMC1, SUMOylation activity) and shows that SLF1 and PML bodies are dispensable for silencing. Furthermore, the authors show that LT can overcome silencing, likely by directly binding to (but not degrading) SMC5/6.Weaknesses:(1) Many of the findings were expected based on recent publications.

There have been no manuscripts describing the role of SIMC1-SLF2 in ecDNA silencing. There have been studies describing SLF2’s roles in ecDNA silencing, but these suggested SLF2 had an SLF1 independent role, with no mention of an alternate Nse5-like cofactor. Our earlier study in eLife (1) described the identification of SIMC1 as an Nse5-like cofactor for SLF2 but did not test potential roles of the complex in ecDNA silencing. Also, the apparent dispensability of PML NBs in plasmid silencing (in U2OS cells) was unexpected based on recent publications. Finally, SV40 LT has not previously been implicated in SMC5/6 inhibition, which may occur through novel mechanisms.

(2) While the data are consistent with SIMC1 playing the main function in plasmid silencing, it is possible that SLF1 contributes to silencing, especially in the absence of SIMC1. This would potentially explain the discrepancy with the data reported in ref. 50. SLF2 deletion has a stronger effect on expression than SIMC1 deletion in many but not all experiments reported in this manuscript. A double mutant/deletion experiments would be useful to explore this possibility.

It is interesting to note that the data in ref. 50 (11) is also at odds with that in ref. 45 (8) in terms of defining a role for SLF1 in the silencing of unintegrated HIV-1 DNA. The Irwan study showed that SLF1 deficient cells exhibit increased expression of a reporter gene from unintegrated HIV-1, whereas the Dupont study found that SLF1 deletion, unlike SLF2 deletion, has no effect. It is unclear what the basis of this discrepancy is. In line with the Dupont study, we found no effect of SLF1 deletion on plasmid expression (Figure 4B), whereas SLF2 deletion increased reporter expression (Figure 3A/B). It is possible that SLF1 could support some plasmid silencing in the absence of SIMC1, especially considering the gross structural similarity in their C-terminal Nse5-like domains. However, we have been unable to generate double-knockout SIMC1 and SLF1 cells to test such a possibility, and shSLF1 has been ineffective.

(3) SLF2 is part of both types of loaders, while SLF1 and SIMC1 are specific to their respective loaders. Did the authors observe differences in phenotypes (growth, sensitivities to DNA damage) when comparing the mutant cell lines or their construction? This should be stated in the manuscript.

We have not observed significant differences in the growth rates of each cell line, and DNA damage sensitivities are as yet untested.

(4) It would be desirable to have control reporter constructs located on the chromosome for several experiments, including the SUMOylation inhibition (Figures 5A and 5-S2) and LT expression (Figure 5D) to exclude more general effects on gene expression.

We have repeated all GFP reporter assays using integrated versus episomal plasmid DNA. A seminal study by Decorsière et al. (6) showed that SMC5/6 degradation by HBx of HBV increased transcription of episomal but not chromosomally integrated reporters. In line with this data, the deletion of SLF2 does not notably impact the expression of our GFP reporter construct when it is genomically integrated (Figure 3—figure supplement 1C).

Somewhat surprisingly, given the generally transcriptionally repressive roles of SUMO, inhibition of the SUMO pathway with SUMOi did not significantly impact the expression of our genomically integrated GFP reporter, versus the episomal plasmid (Figure 5—figure supplement 1C). Finally, the expression of SV40 LT, which enhances plasmid reporter expression (Figure 5D), also did not notably affect expression of the same reporter when located in the genome (Figure 5—figure supplement 3B). This is an interesting result, which is in line with an early study showing that HBx of HBV induces transcription from episomal, but not chromosomally integrated reporters (12). This further suggests that SV40 LT acts similarly to other early viral proteins like HBx and Vpr to counteract or bypass SMC5/6 restriction, amongst their multifaceted functions. Clearly, further analyses are needed to define mechanisms of LT in counteracting SMC5/6, but they do not appear to include complex degradation as seen with HBx and Vpr.

(5) Figure 5A: There appears to be an increase in GFP in the SLF2-/- cells with SUMOi? Is this a significant increase?

No significant difference was found between WT, SIMC1-/- or SLF2-/- when treated with SUMOi (p>0.05). The p-value is 0.0857 (when comparing SLF2-/- to WT in the SUMOi condition) This is described in the figure legend to Figure 5.

(6) The expression level of SFL2 mut1 should be tested (Figure 3B).

Full length SLF2 (WT or mutants) has been undetectable by western analyses. However, truncated SLF2 mut1 expresses well and binds SIMC1 but not SMC6 (Figure 1C). Moreover, full length SLF2 mut1 expression was confirmed by qPCR – showing a somewhat higher expression level than SLF2 WT (Figure 3—figure supplement 1B).

**Reviewer #1 (Recommendations for the authors):**
There are some points about the presented data that need to be clarified.(1) Figures 3, 4B, and 5. The authors should rule out the possibility that the reported effects on transcription were due to alterations in plasmid number. This is particularly important, taking into account the importance of SMC5/6 in DNA replication.

We used qPCR to assess plasmid copy number versus genomic DNA in our cell lines, testing at 72 hours post transfection to avoid any impact of cytosolic DNA (13). Our qPCR data show that there is no significant impact on plasmid copy number across our cell lines i.e. WT and SLF2 null. SMC5/6 has a positive role in DNA replication progression on the genome (e.g. (14)), so loss of SMC5/6 “targeting” in SIMC1 and SLF2 null cells would be unlikely to promote replication fork progression per se.

(2) Figure S1A. In contrast to the statement in the text, the SIMC1-combo control is affected in its binding to SLF2; however, it is not affected in its binding to SMC6. This is somehow unexpected because it suggests that the solenoid-like structure is not required for SMC6 binding, just specific patches at either SIMC or SLF2. This should be commented on.

We appreciate the reviewer’s observation regarding the discrepancy between Figure S1A and the text. This was our oversight. The data show that SLF2 recovery was reduced in the pull-down with the SIMC1 combo control mutant, while SLF2 expression was unchanged. Because SLF2 or SIMC1 variants that fail to associate typically show poor expression (1), these findings suggest that the SIMC1 combo control mutant associates with SLF2, albeit more weakly. Since the mutations were introduced into surface residues of SIMC1, it is not immediately clear how they would weaken the interaction or destabilize the complex. In contrast, SMC6 was fully recovered with the SIMC1 combo control mutant, indicating that the SIMC1–SMC6 interaction remains stable without stoichiometric SLF2. This may reflect direct recognition of a SIMC1 binding epitope or stabilization of its solenoid structure by SMC6, although this interpretation remains uncertain given the unstable nature of free SIMC1 and SLF2. Alternatively, SMC6 may have co-sedimented with the SIMC1 combo control mutant together with SLF2, which was initially retained but subsequently lost during washing, whereas SMC6 remained due to its limited solubility in the absence of other SMC5/6 subunits. While further mechanistic analysis will require purified SMC5/6 components, our data support the AlphaFold-based model by demonstrating that SIMC1 mutations on the non–SMC6-contacting surface retain association with SMC6. The text has been revised accordingly.

(3) The SLF2-only mutant has alterations that affect interactions with both SLF2 and SIMC1. Is it not another Mixed mutant?

We appreciate the reviewer’s observation regarding the discrepancy between the mutant name (“SLF2only”) and its description (“while N947 forms salt bridges with SIMC1”). The previous statement was inaccurate due to a misinterpretation of several AlphaFold models. Across these models, the SIMC1– SLF2 interface residues remain largely consistent, but the SIMC1 residue R470 exhibits positional variability—contacting N947 in some models but not in others. Given this variability and the absence of an experimental structure, we have revised the text to avoid overinterpretation. Because the N947 side chain is oriented toward SLF2 and consistently forms polar contacts with the H1148 side chain and G1149 backbone, we have renamed this mutant “SLF2-facing,” which more accurately describes its modeled environment. The other mutants are likewise renamed “SIMC1-facing” and “SIMC1–SLF2groove-facing,” providing a clearer and more consistent description of the interface.

(4) The SLF2-only mutant still displays clear interactions with SMC6. Can this be explained with the AlphaFold model?

SIMC1 may contribute more substantially to SMC6 binding than SLF2, consistent with our mutagenesis results. However, the energetic contributions of individual residues or proteins cannot be quantitatively inferred from structural models alone. Comprehensive experimental and computational analyses would be required to address this point.

(5) The conclusions about the role of SUMOylation are vague; it is already known that its general effect on transcription repression, and the authors already demonstrated that SIMC interacts with SUMO pathway factors. Concerning the epistatic effect, the experiment should be done at a lower inhibitor concentration; at 100 nM there is not much margin to augment according to the kinetics analysis in Figure S5.

The SUMO pathway is indeed thought to be generally repressive for transcription. Notably, in response to a suggestion from Reviewer 3 (public review point 4), we have repeated several of our GFP expression assays using cells with the GFP reporter plasmid integrated into the genome (please see Figure 3—figure supplement 1C; Figure 5—figure supplement 1C; Figure 5—figure supplement 3B). This type of integrated reporter does not show elevated expression following inhibition of the SMC5/6 complex, unlike ecDNAs (6,10). Interestingly, SUMOi, LT expression, and SLF2 knockout also did not notably impact the expression of our integrated GFP reporter Figure 3—figure supplement 1C; Figure 5—figure supplement 1C; Figure 5—figure supplement 3B, unlike that of the plasmid (ecDNA) reporter. Given the “general” inhibitory effect of SUMO on transcription, the SUMOi result was not expected, and it opens further interesting avenues for study.

In Figure 5—figure supplement 1A, 100 nM SUMOi increases reporter expression well below the highest SUMOi dose. We believe that the ~3-4 fold induction of GFP expression in SLF2 null cells, if independent of SUMOylation, should further increase GFP expression. The impact of SUMOylation on GFP reporter expression remains “vague”, but our data indicate that SMC5/6 operates within SUMO’s “umbrella” function and provides a starting point for more mechanistic dissection.

(6) Figure 5C. Why is the size different between Input versus GFP-PD?

Please see our response to this question above: reviewer 2, point (5)

**Reviewer #2 (Recommendations for the authors):**
If further data could be provided to extend on that which is presented, then publication as a 'standalone research article' may be appropriate, but not in its present form.

We submitted this manuscript as a “Research Advance” not as a standalone research article, given that it was an extension of our previous research article (1).

**Reviewer #3 (Recommendations for the authors):**
(1) The term 'LT' should be defined in the title

We have updated the title accordingly.

(2) This reviewer found the nomenclature of the SMC6 mutants confusing (SIMC1-only...). Either rephrase or define more clearly in the text and the figures.

We agree with the reviewer and have renamed the mutants as “SIMC1-facing”, “SLF2-facing,”, and “SIMC1–SLF2-groove-facing”.

(3) The authors could better emphasize that LT blocks silencing in trans (not only on its cognate target sequence in cis). This is consistent with the observed direct binding to SMC5/6.

We appreciate the suggestion to further emphasize the impact of LT on plasmid silencing. We did not want to overstate its impact at this time because we do not know if it directly binds SMC5/6 or indeed affects SMC5/6 function more broadly. LT expression like HBx, does cause induction of a DNA damage response, but we cannot at this point tie that response to SMC5/6 inhibition alone.

(4) Figure 5 S1: the merge looks drastically different. Is DAPI omitted in the wt merge image?

Thank you for noting this issue. We have corrected the image, which was impacted by the use of an underexposed DAPI image.

(5) Figure 1: how is the structure in B oriented relative to A? A visual guide would be helpful.

We have added arrows to indicate the view orientation and rotational direction to turn A to B.

(6) Line 126, unclear what "specificity" here means.

We have revised the sentence without this word, which now starts with “To confirm the SIMC1-SMC6 interface, we introduced….”

(7) Line 152, The statement implies that the conserved residues are needed for loader subunits interactions ('mediating the SIMC1-SLF2 interaction"). Does Figure 1C not show that the residues are not important? Please clarify.

Thank you for noting this writing error. We have corrected the sentence to provide the intended meaning. It now reads "Collectively, these results confirm that the conserved surface patch of SIMC1SLF2 is essential for SMC6 binding.”

References

(1) Oravcova M, Nie M, Zilio N, Maeda S, Jami-Alahmadi Y, Lazzerini-Denchi E, Wohlschlegel JA, Ulrich HD, Otomo T, Boddy MN. The Nse5/6-like SIMC1-SLF2 complex localizes SMC5/6 to viral replication centers. Elife. 2022;11. PMCID: PMC9708086

(2) Sullivan CS, Pipas JM. T antigens of simian virus 40: molecular chaperones for viral replication and tumorigenesis. Microbiol Mol Biol Rev. 2002;66(2):179-202. PMCID: PMC120785

(3) Gilinger G, Alwine JC. Transcriptional activation by simian virus 40 large T antigen: requirements for simple promoter structures containing either TATA or initiator elements with variable upstream factor binding sites. J Virol. 1993;67(11):6682-8. PMCID: PMC238107

(4) Qadri I, Conaway JW, Conaway RC, Schaack J, Siddiqui A. Hepatitis B virus transactivator protein, HBx, associates with the components of TFIIH and stimulates the DNA helicase activity of TFIIH. Proc Natl Acad Sci U S A. 1996;93(20):10578-83. PMCID: PMC38195

(5) Aufiero B, Schneider RJ. The hepatitis B virus X-gene product trans-activates both RNA polymerase II and III promoters. EMBO J. 1990;9(2):497-504. PMCID: PMC551692

(6) Decorsiere A, Mueller H, van Breugel PC, Abdul F, Gerossier L, Beran RK, Livingston CM, Niu C, Fletcher SP, Hantz O, Strubin M. Hepatitis B virus X protein identifies the Smc5/6 complex as a host restriction factor. Nature. 2016;531(7594):386-9.

(7) Murphy CM, Xu Y, Li F, Nio K, Reszka-Blanco N, Li X, Wu Y, Yu Y, Xiong Y, Su L. Hepatitis B Virus X Protein Promotes Degradation of SMC5/6 to Enhance HBV Replication. Cell Rep. 2016;16(11):2846-54. PMCID: PMC5078993

(8) Dupont L, Bloor S, Williamson JC, Cuesta SM, Shah R, Teixeira-Silva A, Naamati A, Greenwood EJD, Sarafianos SG, Matheson NJ, Lehner PJ. The SMC5/6 complex compacts and silences unintegrated HIV-1 DNA and is antagonized by Vpr. Cell Host Microbe. 2021;29(5):792-805 e6. PMCID: PMC8118623

(9) Felzien LK, Woffendin C, Hottiger MO, Subbramanian RA, Cohen EA, Nabel GJ. HIV transcriptional activation by the accessory protein, VPR, is mediated by the p300 co-activator. Proc Natl Acad Sci U S A. 1998;95(9):5281-6. PMCID: PMC20252

(10) Diman A, Panis G, Castrogiovanni C, Prados J, Baechler B, Strubin M. Human Smc5/6 recognises transcription-generated positive DNA supercoils. Nat Commun. 2024;15(1):7805. PMCID: PMC11379904

(11) Irwan ID, Bogerd HP, Cullen BR. Epigenetic silencing by the SMC5/6 complex mediates HIV-1 latency. Nat Microbiol. 2022;7(12):2101-13. PMCID: PMC9712108

(12) van Breugel PC, Robert EI, Mueller H, Decorsiere A, Zoulim F, Hantz O, Strubin M. Hepatitis B virus X protein stimulates gene expression selectively from extrachromosomal DNA templates. Hepatology. 2012;56(6):2116-24.

(13) Lechardeur D, Sohn KJ, Haardt M, Joshi PB, Monck M, Graham RW, Beatty B, Squire J, O'Brodovich H, Lukacs GL. Metabolic instability of plasmid DNA in the cytosol: a potential barrier to gene transfer. Gene Ther. 1999;6(4):482-97.

(14) Gallego-Paez LM, Tanaka H, Bando M, Takahashi M, Nozaki N, Nakato R, Shirahige K, Hirota T. Smc5/6-mediated regulation of replication progression contributes to chromosome assembly during mitosis in human cells. Mol Biol Cell. 2014;25(2):302-17. PMCID: PMC3890350